

# Higher-form symmetries,
# anomalous magnetohydrodynamics, and holography

Arpit Das[1⋆], Ruth Gregory[2†] and Nabil Iqbal[1‡]

**1** Centre for Particle Theory, Department of Mathematical Sciences,
Durham University, South Road, Durham DH1 3LE, UK
**2** Department of Physics, King's College London, The Strand, London, WC2R 2LS, UK

⋆ arpit.das@durham.ac.uk , † ruth.gregory@kcl.ac.uk ,
‡ nabil.iqbal@durham.ac.uk

## Abstract

In $U(1)$ Abelian gauge theory coupled to fermions, the non-conservation of the axial current due to the chiral anomaly is given by a dynamical operator $F_{\mu\nu}\tilde{F}^{\mu\nu}$ constructed from the field-strength tensor. We attempt to describe this physics in a universal manner by casting this operator in terms of the 2-form current for the 1-form symmetry associated with magnetic flux conservation. We construct a holographic dual with this symmetry breaking pattern and study some aspects of finite temperature anomalous magnetohydrodynamics. We explicitly calculate the charge susceptibility and the axial charge relaxation rate as a function of temperature and magnetic field and compare to recent lattice results. At small magnetic fields we find agreement with elementary hydrodynamics weakly coupled to an electrodynamic sector, but we find deviations at larger fields.

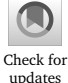

# 1 Introduction

In this work we will discuss the finite temperature physics of a magnetohydrodynamic *chiral* plasma, i.e. an electrodynamic plasma with an axial $U(1)_A$ current $j_A$ that is afflicted by an Adler-Bell-Jackiw anomaly:

$$\partial_\mu j_A^\mu = -\frac{1}{16\pi^2}\epsilon^{\mu\nu\rho\sigma}f_{\mu\nu}f_{\rho\sigma}\,. \tag{1}$$

Here it is understood that this expression arises in a theory of dynamical electromagnetism, and $f_{\mu\nu}$ is the field strength of this fluctuating gauge field. We stress that the non-conservation of the axial current is given by a *dynamical* operator. While the analysis presented here will be from a more formal, holographic, perspective, the system has clear phenomenological interest, with applications to baryon number violation [1–5], primordial magnetic fields [6–8], magnetised baryogenesis [9–11], and Dirac and Weyl semi-metals in condensed matter systems (for a review see [12]).

As described in [1], a quantity of interest in $U(1)$ anomalous processes is the relaxation rate of the chiral charge density $j_A^0$. This is what we seek to compute in this work. In our opinion a fully universal hydrodynamic treatment of this problem has not yet been given; indeed it is somewhat unclear whether one should exist.

We begin by carefully stating the problem and distinguishing it from the large existing literature on anomalous hydrodynamics. To orient ourselves, it is helpful to first imagine a weakly-coupled realization of the physics that we are interested in. Consider the following Lagrangian describing a massless Dirac fermion coupled to dynamical electromagnetism with

photon $a$:

$$S[a, \psi] = \int d^4x \left( -\frac{1}{4e^2} f^2 + \bar{\psi} \left( \slashed{\partial} - i \slashed{a} \right) \psi \right). \tag{2}$$

We will be interested in placing this system at finite temperature and understanding the hydrodynamic description.

What are the global symmetries of this system? It has a 1-form $U(1)^{(1)}$-symmetry associated with the conservation of magnetic flux:

$$\partial_\mu J^{\mu\nu} = 0, \qquad J^{\mu\nu} \equiv \frac{1}{2} \epsilon^{\mu\nu\rho\sigma} f_{\rho\sigma}. \tag{3}$$

In this work we assume the reader is familiar with higher-form symmetries [13] and we will often denote a $p$-form $U(1)$-symmetry by $U(1)^{(p)}$. A recent review of applications of higher-form symmetries can be found in [14].

The symmetry associated with vector phase rotations of the $\psi$ field $\psi \to e^{i\alpha}\psi$ is gauged and does not correspond to a global symmetry. Classically, the theory appears to have a conventional (i.e. 0-form) $U(1)_A^{(0)}$-symmetry associated with $\psi \to e^{i\alpha\gamma^5}\psi$; however at the quantum mechanical level the conservation of the associated current is broken by the Adler-Bell-Jackiw anomaly (1). Note that the right-hand side of this expression is an *operator*, as $f$ is a fluctuating dynamical field. This should be contrasted with the case of a 't-Hooft anomaly, where the right-hand side of a current-conservation equation involves a fixed external source that can be set to zero.

We are interested in understanding the realization of the symmetries at finite temperature. This is the domain of *hydrodynamics*, which describes how conserved quantities relax towards thermal equilibrium. Hydrodynamics in the presence of a 't-Hooft anomaly is a rich and well-studied field [15, 16] (see also [12]). The situation with the anomaly above is somewhat different. As the right-hand side is a fluctuating operator, there is no longer a strictly universal sense in which the axial current is conserved. Thus it appears that the only true global symmetry of the system is the 1-form symmetry (3). A naive application of the conventional formalism of hydrodynamics applied to this system would then only involve a study of the 1-form symmetry in thermal equilibrium. Such an analysis was performed in [17], where it was shown that the resulting framework is essentially a reformulation of conventional relativistic magnetohydrodynamics, i.e. a description of an electrodynamic plasma (a holographic description of this plasma from this point of view was given in [18, 19]). The only conserved quantity here is the usual magnetic flux, and this description of course makes no reference to the axial current whatsoever.

Nevertheless, to us this situation seems somewhat unsatisfactory; after all, from an applied viewpoint, it seems clear that the finite-temperature dynamics of (2) has a rich and physically relevant phenomenology. This physics is usually accessed by coupling the equations of (ungauged) hydrodynamics with a 't Hooft anomaly to weakly coupled electrodynamics "by hand" [7, 20]; in particular see [21] which constructs a hydrodynamic theory in a formal expansion in the anomaly coefficient. These constructions are not fully universal, and the domain of validity of the resulting theories is not entirely clear. In particular, recent work on the lattice [1, 2] that computes the charge relaxation rate $\Gamma_A$ shows a disagreement with the predictions of the above hydrodynamic theories, suggesting that short-distance fluctuations play an important role that is not captured by the non-universal theories above.

It is difficult to come up with a universal hydrodynamic theory for this model. In fact, towards the end of our analysis, we will see that we have reasons to believe that such a universal hydrodynamic description might not exist for such a set up. For now we will explore this problem in a new way, by using holographic duality to explore aspects of the finite-temperature dynamics of a system in the same universality class as the weakly coupled theory described

above. To construct our holographic dual theory, we must first carefully understand the symmetries in a manner that is independent of the description. One route to understand this is to note that the right-hand side of the expression above can be written in terms of the current $J^{\mu\nu}$ for the 1-form symmetry:

$$\partial_\mu j_A^\mu = k\,\epsilon_{\mu\nu\rho\sigma}J^{\mu\nu}J^{\rho\sigma}\,, \qquad k \equiv \frac{1}{16\pi^2}\,. \tag{4}$$

We may now try to describe the dynamics of a system with a conserved 2-form current $J^{\mu\nu}$ and a 1-form axial current $j_A^\mu$ that satisfies the above non-conservation equation; this may be thought of as a kind of intertwining of the (genuine) 1-form symmetry and the (broken by the anomaly) 0-form symmetry. (To our knowledge, this particular intertwining does not appear to have a precise universal characterization in the field theory literature; however see further discussion in the conclusion).[1]

In this work we will first construct a holographic model that possesses the above symmetries. We will then perform a preliminary investigation of the resulting holographic system. In particular, we study the system in the presence of a background magnetic field. We will explicitly compute the charge relaxation rate in this model and compare it both to elementary hydrodynamics with weakly coupled electromagnetism and to recent lattice results; we will find agreement with hydrodynamics at low magnetic fields, but disagreement at large magnetic fields; this suggests that UV fluctuations are important for a quantitative determination of this relaxation rate.

A short outline of the rest of this paper is as follows. In Section 2 we review a simple hydrodynamic discussion of the charge relaxation rate. In Section 3 we introduce the holographic model that we will study in the remainder of the paper. In Section 4 we place this model at finite temperature and study some aspects of static response (i.e. the analogue of the charge susceptibility). In Sections 5 and 6 we study finite-frequency response (both analytically at small frequencies and numerically) and we conclude with a brief discussion in Section 7.

## 2 Hydrodynamic calculation of relaxation rate

We begin our study by defining the relaxation rate that we will compute and using elementary physical arguments to understand what may control it; at the end we will compare the resulting physics to our holographic construction.

We first review the usual hydrodynamic computation of this charge relaxation rate. This is done in the usual framework of "chiral MHD". As described above, this means we assume a certain anomalous contribution to the dynamical electric current and couple it perturbatively to an MHD sector. We particularly highlight [21], where the authors perform a hydrodynamic study where the anomaly coefficient $k$ is treated perturbatively; they compute the chiral charge relaxation rate $\Gamma_A$ for small $k$. In this limit, they find that, $\Gamma_A \sim k^2 B^2$ with $B$ being the magnetic field.

We review a similar calculation below: this is physically instructive but as discussed above is not truly universal. This calculation is a very slight generalization of the one presented in [1].

In our notation the anomaly takes the form

$$\partial_\mu j_A^\mu = -2k F_{\mu\nu}\tilde{F}^{\mu\nu} = k\epsilon_{\mu\nu\rho\sigma}J^{\mu\nu}J^{\rho\sigma}\,, \tag{5}$$

---

[1]Note added: after the first version of this paper was released on the arXiv, [22,23] appeared: these works present a precise field-theoretical characterization of the ABJ anomaly in terms of non-invertible symmetries.

where $j_A^\mu$ is the chiral current, $\tilde{F}_{\mu\nu}$ is the Hodge dual of the field strength $F_{\mu\nu}$. (In the case of a single Dirac fermion studied in [1] we have $2k = -\frac{e^2}{8\pi^2}$, where $e$ is the electromagnetic coupling.)

For the homogeneous case (see [1]), we have $\vec{j}_A = 0$ and thus the anomaly equation becomes (for $j_A^0 \equiv \rho = \chi\mu_A$),

$$\chi\frac{d\mu_A}{dt} = -\frac{8k}{V}\int d^3x\ \vec{E}\cdot\vec{B}, \quad (\text{as, } F_{\mu\nu}\tilde{F}^{\mu\nu} = 4\vec{E}\cdot\vec{B}), \tag{6}$$

where $\mu_A$ is the space-independent axial chemical potential, $\rho$ the axial charge density, and $\chi$ the axial charge susceptibility, which in principle could depend on the temperature and background magnetic field.

Thus, we have the chiral relaxation rate as,

$$\frac{d\mu_A}{dt} = -\frac{8k}{\chi V}\int d^3x\ \vec{E}\cdot\vec{B}. \tag{7}$$

Following [1] we give below the chiral MHD equations as,

$$\frac{\partial\vec{B}}{\partial t} = \nabla\times\vec{E}, \qquad \frac{\partial\vec{E}}{\partial t} + \nabla\times\vec{B} = -\sigma\vec{E} + 8k\mu_A\vec{B}, \tag{8}$$

where $\sigma$ is the electric conductivity of the plasma, and we have assumed that the density of electric charge is zero, and the plasma has zero velocity. The last term: $8k\mu_A\vec{B}$, is the contribution from the chiral magnetic effect (CME).[2] This system of equations is complemented by the anomaly equation above (7).

Now if we neglect the time-derivative of $\vec{E}$ in (8) we can express $\vec{E}$ in terms of $\vec{B}$ using (8). Then, we get for long-range fluctuations of the gauge fields (that is $\nabla\times\vec{B}\to 0$),

$$\frac{d\mu_A}{dt} = -\frac{8k}{\sigma\chi V}\int d^3x\ \left(8k\mu_A\vec{B}\right)\cdot\vec{B} = -\frac{64k^2B^2}{\sigma\chi}\mu_A \equiv -\Gamma_A\mu_A, \tag{9}$$

where in the last equality $\Gamma_A$ is defined as the rate of chirality non-conservation in the presence of an external homogeneous magnetic field $\vec{B}$. The solution of Eq.(9) goes as,

$$\mu_A(t) = e^{-\Gamma_A t}\mu_{A,0} \quad (\text{where } \mu_{A,0} \text{ is an integration constant}). \tag{10}$$

From Eq.(9) we see that,

$$\Gamma_A = \frac{64k^2B^2}{\sigma\chi}, \tag{11}$$

i.e. the relaxation rate is quadratic in the magnetic field. We will compare this elementary discussion to an explicit holographic calculation later.

We note that in [1,2], the authors perform a numerical lattice computation in determining the chiral charge relaxation rate $\Gamma_A$. They also found it to be quadratic in the magnetic field $B$. As mentioned above, it was observed that the pre-factor in $\Gamma_A\sim B^2$ is approximately 10 times that of the theoretical predictions of the same pre-factor from hydrodynamics. Calculating this pre-factor in a strongly coupled yet solvable holographic model and comparing it to the existing literature serves as a pragmatic motivation for this study.

---

[2]Note that there is a factor of 2 difference in the CME between this paper and that of [1]. This is owing to the fact that in [1], $\mu_A$ couples to $\frac{j_A^0}{2}$ while here in the definition of $\mu_A$, we have chosen it to couple to $j_A^0$.

# 3 Overview of holographic model

In this section, we will present a bulk holographic theory which realizes the pattern of symmetry non-conservation in (4). We will begin by presenting the bulk action and demonstrating the deformed Ward identity; in the next section, we will describe how we arrived at this theory from dualizing a different bulk action. (For a summary of our conventions and notation see Appendix A.)

## 3.1 Holographic bulk action

We desire a bulk theory with the following properties: it should have a bulk massless 2-form $B_{MN}$; as explained in detail in [18], this is associated with a global 1-form symmetry in the boundary, as $B_{MN}$ is dual to the boundary 2-form current $J^{\mu\nu}$. The action should also have a vector field which we call $E_M$. $E_M$ is dual to a vector operator representing the axial current $j_A^\mu$ on the boundary; this vector operator should be understood as the axial current, and it is not conserved. Thus, $E_M$ should not enjoy a bulk gauge symmetry. (We will see in a later section that $E_M$ is of the form $A_M - \partial_M \phi$, where $A_M$ and $\phi$ enjoy (bulk) gauge symmetries in such a way that $E_M$ is gauge-invariant). However, the divergence of $j_A^\mu$ on the boundary is not completely unconstrained; rather its divergence should be related to the a double-trace operator of the 2-form current $J_{\mu\nu}$ by the following anomaly equation

$$\partial_\mu j_A^\mu = k\, \epsilon_{\mu\nu\rho\sigma} J^{\mu\nu} J^{\rho\sigma}\,, \tag{12}$$

where $k$ is a parameter that should enter the bulk action.

We now present a bulk action which satisfies the above properties:[3]

$$S[E,B] = \int d^5x \sqrt{-g}\left[-\frac{1}{4}G^2 - \frac{1}{12}H^2 + 16k^2(E\cdot H)^2 - \frac{k}{3}\epsilon_{PQRMN}H^{PQR}E_L H^{LMN}\right]. \tag{13}$$

Here $G = dE$ and $H = dB$ are the field strengths of $E$ and $B$ respectively, and we have defined $H^2 = H^{PQR}H_{PQR}$, $(E\cdot H)^2 = E_L H^{LMN} E^P H_{PMN}$. The theory has an invariance under a 1-form gauge symmetry:

$$B \to B + d\Lambda\,, \tag{14}$$

with $\Lambda$ an arbitrary 1-form. $E$ clearly enjoys no explicit gauge symmetry; note however that the "mass" terms for $E$ have a specific structure, involving couplings to $H$ that are parameterized by a single coupling $k$. We will show that this structure encodes the anomaly (12). This action should be understood as being correct to order $\mathcal{O}(E^2)$; as we will show, the anomaly structure (12) above is only correctly represented to that order. Below we will also present an algorithm that can be used to obtain an action that is correct to all orders in $E$, though we will not require it for our purposes.

Motivated by studies pertaining to similar anomalies, a bulk action involving anomaly-inspired mass terms for gauge fields was studied in [24] (see Eq.(30) in [24]). There, the anomaly is thought to be sourced by a dynamical non-Abelian gauge field, which does not have an associated 1-form symmetry. In our case, the dynamical gauge field is Abelian, and thus the non-conservation of the current is precisely related to a 2-form current with universal dynamics; thus our action takes a more constrained form (and describes somewhat different physics) compared to that in [24].[4]

---

[3]In the action below one can certainly have higher order terms consistent with the symmetry structure described above but they won't contribute to the holographic calculation which we describe at linear order in $k$.

[4]We note that [24] also consider external vector Abelian magnetic fields; however these fields act as sources and are non-dynamical.

The variation of the action above with respect to $B$ results in the following equations of motion:

$$-\frac{1}{2}\partial_L\left(\sqrt{-g}H^{LMN}\right)-k\,\partial_L\left[\sqrt{-g}\;\epsilon^{LMNQR}E^P H_{PQR}\right]$$
$$-\frac{k}{3}\partial_L\left[\sqrt{-g}\;H_{PQR}\left(E^L\epsilon^{PQRMN}+E^M\epsilon^{PQRNL}+E^N\epsilon^{PQRLM}\right)\right]=0,\tag{15}$$

and similarly, we have the following equations for the variation with respect to $E$:

$$\frac{1}{\sqrt{-g}}\partial_M\left(\sqrt{-g}G^{ML}\right)=-32k^2H^{LQR}E^P H_{PQR}+\frac{k}{3}\epsilon_{PQRMN}H^{PQR}H^{LMN}.\tag{16}$$

Note that if $k=0$ these equations of motion decouple into free Maxwell equations for $E$ and $B$ respectively. Here we work only to linearized order in $E$.

We would now like to interpret this bulk physics holographically. We begin with the 2-form $B$. As usual, we may construct the boundary 2-form current $J^{\mu\nu}$ by varying the action with respect to the boundary value of $B_{\mu\nu}$ (see appendix A for convention regarding the boundary current defined below)

$$J^{\mu\nu}(x)=2\frac{\delta S}{\delta B_{\mu\nu}(\infty)}.\tag{17}$$

The on-shell variation of the boundary value of the action may be reduced to a variation with respect to the radial derivative $\partial_r B_{\mu\nu}$, and we thus find

$$J^{\rho\sigma}(x)=2\lim_{r\to\infty}\frac{\delta S}{\delta\left(\partial_r\left(B_{\rho\sigma}\right)\right)}\tag{18}$$

$$=\lim_{r\to\infty}\sqrt{-g}\left[-H^{r\rho\sigma}-2k\,\epsilon^{r\rho\sigma\mu\nu}E^\alpha H_{\alpha\mu\nu}-\frac{2k}{3}\left[H_{\alpha\beta\gamma}\{E^r\epsilon^{\alpha\beta\gamma\rho\sigma}+E^\rho\epsilon^{\alpha\beta\gamma\sigma r}+E^\sigma\epsilon^{\alpha\beta\gamma r\rho}\}\right]\right].\tag{19}$$

Here the first term is standard [18]; the others arise from the physics associated with the anomaly. We have omitted terms of order $\mathcal{O}(E^2)$; this is because in this work we will study only the linearized equations of motion of $E$.

Note that the radial component of the bulk wave equation (15) for $B_2$ ensures that we have

$$\partial_\mu J^{\mu\nu}(x)=0,\tag{20}$$

i.e. that the 2-form current is conserved.

We now turn to $E$. We may construct the dual current as in (17):

$$j_A^\mu=\frac{\delta S}{\delta E_\mu(\infty)}=-\sqrt{-g}G^{r\mu}(r\to\infty).\tag{21}$$

The last expression is standard for the boundary operator dual to a vector field.

Let us now understand the non-conservation of $j_A^\mu$, i.e. let us derive (12) holographically at linear order in $\mathcal{O}(E)$. From the above expression for the 2-form current at the boundary let us now compute[5] $k\,J\wedge_4 J$:

$$\left[-\frac{k}{4}\epsilon_{\alpha\beta\mu\nu}J^{\alpha\beta}J^{\mu\nu}\right]=\left(\sqrt{-g}\right)^2\left(\frac{8k^2}{\sqrt{-g}}E^\alpha H_{\alpha\mu\nu}H^{r\mu\nu}-\frac{k}{4}\epsilon_{\alpha\beta\mu\nu}H^{r\alpha\beta}H^{r\mu\nu}\right).\tag{22}$$

(Note the explicit appearance of factors of the determinant of the metric; this arises from the fact that from the point of view of the bulk $J^{\alpha\beta}$ is not a tensor, as can be seen from its definition in (18).) Now let us consider $L=r$ in (16),

$$32k^2H^{r\mu\nu}E^\alpha H_{\alpha\mu\nu}-k\sqrt{-g}\epsilon_{\alpha\beta\mu\nu}H^{r\alpha\beta}H^{r\mu\nu}=-\partial_\sigma G^{\sigma r}\quad(\sigma\neq r).\tag{23}$$

[5]Note, the boundary metric is flat. Thus we have the following expression relating boundary and bulk Levi-Civita tensors, $\epsilon_{rabcd}=\sqrt{-g}\,\epsilon_{abcd}$.

Usually in AdS/CFT this component of the bulk Maxwell equations of motion is a radial constraint that enforces the conservation of the current $j_A^\mu$; here we see that the current is instead not conserved. Using (22) we see that it is equal to

$$\partial_\sigma j_A^\sigma = \partial_\sigma \left( \sqrt{-g} \, G^{\sigma r} \right) = k \epsilon_{\alpha\beta\mu\nu} J^{\alpha\beta} J^{\mu\nu}, \tag{24}$$

i.e. equivalent to (12) as desired.

This shows that this holographic theory is in the correct universality class, by which we mean that it correctly links the non-conservation of the 1-form axial current with a bilinear constructed from the 2-form current $J^{\mu\nu}$. The fact that $E$ has no gauge symmetry at all in the bulk is dual to the fact that its non-conservation is given in terms of a dynamical operator that cannot be turned off. We note also that the intermediate steps are somewhat complicated and rely on the detailed structure of the action (13). The reader who is willing to take this action as a given and is interested only in results can now skip to the next section, where we compute the holographic observables of interest.

In the remainder of this section we describe how we construct this action through bulk Poincaré duality.

## 3.2 Dualizing the action

Our approach to constructing the bulk action is essentially the bulk dual of the operation of "gauging a global $U(1)$ symmetry"; i.e. we begin by considering the very well-studied bulk action [24–26][6] for a theory with two 0-form global symmetries $U(1)_A \times U(1)_V$ with a mixed 't Hooft anomaly between them:

$$S_5 \left[ A_1, V_1 \right] = \int_{\mathcal{M}^5} \left( -\frac{1}{2} F_2 \wedge \star F_2 - \frac{1}{2} G_2 \wedge \star G_2 - 4k A_1 \wedge F_2 \wedge F_2 - \frac{4k}{3} A_1 \wedge G_2 \wedge G_2 \right). \tag{25}$$

Here $A_1$ and $V_1$ are two 1-form potentials which are holographically dual to the 0-form axial and vector currents respectively, and $F_2 = dV_1$ and $G_2 = dA_1$. The action is invariant (up to a boundary term) under the following gauge symmetries:

$$A_1 \to A_1 + d\Lambda_0, \qquad V_1 \to V_1 + d\lambda_0. \tag{26}$$

The boundary variation of the above gauge-transformation is nonzero, and it is well-understood [25] that this means that the dual field theory has a 't Hooft anomaly for the axial current:

$$\partial_\mu j_A^\mu = k \epsilon^{\mu\nu\rho\sigma} \left( F_{\mu\nu} F_{\rho\sigma} + \frac{1}{3} G_{\mu\nu} G_{\rho\sigma} \right), \tag{27}$$

where $F$ and $G$ are the field strengths of the fixed external sources for the vector and axial currents respectively.

We now want to study a field theory where we have "gauged" the global symmetry $U(1)_V$. In this operation the boundary gauge field $V_1$ will become dynamical, and we will thus lose the 0-form symmetry $U(1)_V$. However we expect to obtain a new 1-form symmetry (and 2-form current) associated with the conserved magnetic flux in our new $U(1)$ gauge theory; in holographic duality, we thus expect to obtain a new 2-form bulk field which we call $B_2$.

It is thus very natural to expect that the bulk operation equivalent to "gauging" on the boundary is to perform a *bulk* Poincaré duality on the bulk 1-form potential $V_1$, replacing it with a 2-form $B_2$. Similar operations have a long history in AdS/CFT, and may be viewed as a

---

[6]We also found the exposition in [27, 28] useful.

higher-form generalization of [29]; see [30,31] for applications of such holographic operations in hydrodynamics. Also, see [32,33] for recent work in a similar holographic context.

We now describe the dualization process below, which proceeds essentially as it would in flat space. This is a well-posed operation, but to the best of our knowledge the details are not present in the literature for a non-linear action of the form (25) even in flat space. We will see some interesting wrinkles arising from the presence of the mixed Chern-Simons term.

### 3.2.1 Poincaré dualization

We follow the usual algorithm to dualize $V_1$, as can be found e.g. in Appendix B of [34] (see also [35]). The action does not depend on $V_1$ directly, but only on its field strength $F_2$; it is thus possible to treat $F_2$ as the dynamical variable rather than $V_1$. However we then need to impose its closure $dF_2 = 0$ through the use of a Lagrange multiplier $B_2$.

We construct the parent action $S_{5p}$ by adding the Lagrange multiplier term's action ($S_c$) to the action $S_5$. We get for $S_{5p}$,[7]

$$
S_{5p}\,[A_1, F_2, B_2] = \underbrace{\int_{\mathcal{M}^5} -\frac{1}{2}F_2 \wedge \star F_2 - \frac{1}{2}G_2 \wedge \star G_2 - 4k A_1 \wedge F_2 \wedge F_2 - \frac{4k}{3}A_1 \wedge G_2 \wedge G_2}_{S_5}
$$
$$
+ \underbrace{\int_{\mathcal{M}^5} dB_2 \wedge F_2}_{S_c}\,,
\tag{28}
$$

where,

$$
S_c = \int_{\mathcal{M}^5} dB_2 \wedge F_2 = \int_{\mathcal{M}^5} d(B_2 \wedge F_2) - B_2 \wedge dF_2\,.
\tag{29}
$$

Then, $\delta_{B_2} S_c = 0$ gives $dF_2 = 0$ (closure of $F_2$). Now imposing the equation of motion $\delta_{F_2} S_{5p} = 0$ yields,

$$
\star F_2 = dB_2 - 8k\,(A_1 \wedge F_2)\,.
\tag{30}
$$

The standard procedure is to now solve for $dB_2$ as a function of $F_2$ and eliminate the latter from the action entirely.

### 3.2.2 Gauge symmetries of $S_5$

Before doing so, we discuss the symmetries: note that the realization of the 0-form gauge symmetry associated with $A_1$ has changed, as $F_2$ is now closed only on-shell. In the action as given in (28), consider instead the following gauge transformations,

$$
A_1 \rightarrow A_1 + d\Lambda_0\,, \qquad B_2 \rightarrow B_2 + 8k\,\Lambda_0 F_2\,.
\tag{31}
$$

It is easy to show that with the above gauge transformations, the action $S_{5p}$ as given in (28) is gauge-invariant but the equation of motion (30) is not (off-shell). It fails to be gauge-invariant by a term $8k\Lambda_0\,dF_2$; since $F_2$ is an independent dynamical field now, it is not necessarily a closed 2-form unless we impose $B_2$'s equations of motion. This may appear problematic: the action is gauge-invariant under the gauge transformations but the equations of motion are not off-shell gauge-invariant under the same gauge transformations. We shall remedy this below by introducing a new auxiliary field $\phi_0$. The equations of motion of $\phi_0$ shall serve as

---

[7]In the action $S_{5p}$, $F_2$ is now a dynamical field.

a constraint (which we refer to as a *gauge-invariant constraint*) for imposing the closure of $F_2 \wedge dF_2$.

Alternatively, we can introduce $\phi_0$ by the following argument. Taking a step back, let us first consider variations of the action $S_5$ as given in Eq.(25) w.r.t. the gauge transformations as given in Eq.(26). We have,

$$\delta S_5 = 8k\, \Lambda_0 F_2 \wedge dF_2 + \frac{8k}{3}\, \Lambda_0\, G_2 \wedge dG_2 + \text{boundary terms}, \tag{32}$$

where the first two terms on the LHS vanish owing to the fact that both $F_2$ and $G_2$ in $S_5$ are closed 2-forms. This then ensures the invariance of $S_5$ (up to boundary terms) under gauge transformations (26). However, in the dualized form $S_{5p}$, $F_2$ is an arbitrary 2-form and not necessarily closed. So, due to the non-closure of $F_2$ we now may or may not have $F_2 \wedge dF_2 = 0$ (off-shell). Therefore, in addition to imposing the closure of $F_2$ by a Lagrange multiplier $B_2$ we have to add to $S_{5p}$ another Lagrange multiplier to impose the constraint $F_2 \wedge dF_2 = 0$ (*gauge-invariant constraint*). From the degree of the term $F_2 \wedge dF_2$, it is clear that the Lagrange multiplier in this case would be a 0-form, say $\phi_0$. Furthermore, as $S_{5p}$ has to remain gauge-invariant under $A_1 \to A_1 + d\Lambda_0$, $\phi_0$ has to be a gauge field with its own gauge transformation given as, $\phi_0 \to \phi_0 + \Lambda_0$ (by construction). Then, we have $S_{5p}$ as,

$$S_{5p} = \int_{\mathcal{M}^5} -\frac{1}{2}F_2 \wedge \star F_2 - \frac{1}{2}G_2 \wedge \star G_2 - 4k A_1 \wedge F_2 \wedge F_2 - \frac{4k}{3}A_1 \wedge G_2 \wedge G_2 + dB_2 \wedge F_2$$
$$- \int_{\mathcal{M}^5} 8k\, \phi_0 F_2 \wedge dF_2, \tag{33}$$

which can be re-written as,

$$S_{5p} = \int_{\mathcal{M}^5} -\frac{1}{2}F_2 \wedge \star F_2 - \frac{1}{2}G_2 \wedge \star G_2 - 4k\,(A_1 - d\phi_0) \wedge F_2 \wedge F_2 - \frac{4k}{3}A_1 \wedge G_2 \wedge G_2 + dB_2 \wedge F_2, \tag{34}$$

with $E_1 \equiv A_1 - d\phi_0$ being a vector field. Note that the emergence of the gauge-invariant field $E_1$ is precisely the structure anticipated earlier, which we now see emerges naturally when demanding off-shell gauge-invariance. We also note that the equations of motion of $\phi_0$ are redundant – they follow automatically from the equations of $A_1$.[8]

Now let us give below the gauge transformations of $A_1$ and $\phi_0$,

$$A_1 \to A_1 + d\Lambda_0, \qquad \phi_0 \to \phi_0 + \Lambda_0. \tag{35}$$

With these gauge transformations the $F_2$ equation of motion, $F_2 = -\star[dB_2 - 8k\,(E_1 \wedge F_2)]$, remains gauge-invariant even off-shell, and everything is consistent with the Poincaré dualization procedure. Clearly, $S_{5p}$ is also invariant under (35) up to boundary terms.

The conclusion of the above discussion is that one should be careful while performing Poincaré dualization, as at times one may be required to impose a *gauge-invariant constraint* along with the usual closure constraint.

### 3.2.3 Inverse operation

Let us now proceed to eliminate $F_2$ from the action. We can now invert (30) to get $F_2$ in terms of $E_1$ and $B_2$ as below. See Appendix B for details of this calculation.

$$F_{MN} = -\frac{\tilde{c}_1}{6}\epsilon_{PQRMN}H^{PQR} + 8\tilde{c}_1 k\, E^P H_{PMN} + \frac{64}{3}\tilde{c}_1 k^2\, H^{PQR}E^L \epsilon_{PQRL[M}E_{N]}, \tag{36}$$

---

[8]Another way to understand $\phi_0$ is that $e^{i\phi_0}$ is an operator that is charged under the bulk 0-form "instanton" current $\star_5 F \wedge F$; in a conventional formalism where $F = dA$ this current is conserved identically. However, in this formalism its conservation must be enforced by $\phi_0$'s equations of motion.

where $H_{PQR} = \partial_P B_{QR} + \partial_Q B_{RP} + \partial_R B_{PQ}$ (as, $H_3 = dB_2$) and $\tilde{c}_1 \equiv \frac{1}{1+64k^2E^2}$.

Note that, if in $S_5$ we have $k \to 0$, $F_2$'s equations of motion become $F_2 = -\star dB_2$ and if we take $k \to 0$ in the above equation we get $F_2 = -\star dB_2$. $F_{MN}$ above has been written in such a way so that it is manifestly anti-symmetric.

Now let us substitute Eq.(36) into Eq.(34) to obtain below the full non-linear bulk action,

$$
S_{5p} = \int_{\mathcal{M}^5} \sqrt{-g}\, d^5x \left[ \left\{ -\frac{H^2}{12} - \frac{k}{3}\epsilon_{PQRMN}H^{PQR}E_D H^{DMN} + 16k^2\left( (E \cdot H)^2 - \frac{2}{3}E^2 H^2 \right) \right.\right.
$$
$$
\left. -64k^3 \epsilon_{PQRMN}E^P E_L H^{LQR}E_J H^{JMN} + 256k^4\left( E^2(E \cdot H)^2 - \frac{1}{3}E^4 H^2 \right) \right\} \tilde{c}_1^2
$$
$$
\left. + \left\{ \frac{k}{3}\epsilon^{PQRMN}E_P G_{QR}G_{MN} - \frac{1}{4}G^2 \right\} \right]. \tag{37}
$$

Note that truncating the above action to $\mathcal{O}(E^2)$ results in the quadratic action in Eq.(13) above, which we will use for the remainder of this study, in which we consider only small fluctuations about equilibrium.

Now we give below the full 2-form current $J^{\mu\nu}$ obtained from the action above Eq.(37),

$$
J^{\rho\sigma} = 2 \lim_{r\to\infty} \frac{\delta S_{5p}}{\delta\left(\partial_r B_{\rho\sigma}\right)}
$$
$$
= \lim_{r\to\infty}\sqrt{-g}\left[ -H^{r\rho\sigma} - 2k\epsilon^{r\rho\sigma\mu\nu}E^\eta H_{\eta\mu\nu} - \frac{2k}{3}H_{\alpha\beta\gamma}\left( E^r \epsilon^{\alpha\beta\gamma\rho\sigma} + E^\rho \epsilon^{\alpha\beta\gamma\sigma r} + E^\sigma \epsilon^{\alpha\beta\gamma r\rho} \right) \right.
$$
$$
+ 64\left(k^2 + 32k^4 E^2\right)E_\alpha\left( E^r H^{\alpha\rho\sigma} + E^\rho H^{\alpha\sigma r} + E^\sigma H^{\alpha r\rho} \right) - 8\left(1 + 32k^2 E^2\right)k^2 E^2 H^{r\rho\sigma}
$$
$$
\left. -256k^3 E_\alpha E^\kappa H_{\kappa\beta\gamma}\left( E^r \epsilon^{\alpha\beta\gamma\rho\sigma} + E^\rho \epsilon^{\alpha\beta\gamma\sigma r} + E^\sigma \epsilon^{\alpha\beta\gamma r\rho} \right) \right] \tilde{c}_1^2. \tag{38}
$$

Now one can, in principle, check that the anomaly structure of Eq.(12) can be obtained from the above 2-form current by performing an order-by-order (in $k$) comparison of coefficients on both sides of Eq.(12). We have checked this explicitly to $\mathcal{O}(k^3)$.

## 4 Finite temperature physics: Zero frequency

With the holographic action in hand, we will now study the plasma that is obtained from the realization of these symmetries at finite temperature. To heat up our system, we consider the background metric given by the usual planar black brane background

$$
ds^2 = r^2\left( -f(r)dt^2 + d\vec{x}^2 \right) + \frac{dr^2}{r^2 f(r)}, \tag{39}
$$

where $f(r) = 1 - \left(\frac{r_h}{r}\right)^4$ and where we are working in units where the AdS radius $R = 1$. The Hawking temperature of the black brane is $T = \frac{r_h}{\pi}$. We are interested in the physics in the presence of a background magnetic field in the $z$ direction, i.e. a configuration where $\langle J^{tz} \rangle = -b$. From the holographic dictionary (19), we see that this means that $H^{rtz} \neq 0$; solving the equations of motion (15) we see that the background profile is:

$$
H^{rtz} = \frac{b}{r^3}, \qquad H_{rtz} = -\frac{b}{r}, \qquad E = 0. \tag{40}
$$

Note that, here we are working in the so called *probe limit* – where we neglect the backreaction of the magnetic field onto the geometry. In other words, we assume the above background

profile (Eq.(40)) doesn't affect the metric components given in Eq.(39). This is justified in the high-temperature limit; at low temperatures this backreaction cannot be neglected, and one would replace the background with a magnetic brane solution [36,37]. Now we will begin our study by computing the static axial charge susceptibility $\chi$ in the presence of the background magnetic field. For a theory with a conserved charge, the susceptibility can be defined as:

$$\chi = \frac{\partial \langle j_A^t \rangle}{\partial \mu_A}, \qquad \text{leading to,} \ \langle j_A^t \rangle = \chi \mu_A \ \text{(in the linear regime)}, \tag{41}$$

where $\mu_A$ is the axial chemical potential. In the case of a non-conserved axial current the precise definition of the axial chemical potential as a dynamical hydrodynamic variable is somewhat more subtle (see e.g. [25]), but in thermal equilibrium it can be understood as the value of the axial source $A_t$, which coincides with $E_t$ when all fields are static.

## 4.1 Susceptibility

In this section we shall consider the low frequency limit of Eq.(16) and compute the axial charge susceptibility in this model. From Eq.(16) we find,

$$\delta E_r = \frac{(i\omega r^2)\partial_r (\delta E_t) + 4kbf \partial_r (\delta B_{xy})}{r^2 \omega^2 - 64b^2 k^2 f} \quad \text{(with } L = r \text{ in Eq.(16))}, \tag{42}$$

$$\partial_r^2 (\delta E_t) + i\omega \partial_r (\delta E_r) + \left(\frac{3}{r}\right)\partial_r (\delta E_t) + \left(\frac{3}{r}\right)i\omega \delta E_r = \frac{64b^2 k^2}{r^6 f}\delta E_t - \frac{4kbi\omega}{r^6 f}\delta B_{xy}$$

$$\text{(with } L = t \text{ in Eq.(16))}. \tag{43}$$

Now plugging $\delta E_r$ from Eq.(42) in Eq.(43) and taking the $\omega \to 0$ limit we obtain,[9]

$$\partial_r^2 (\delta E_t) + \left(\frac{3}{r}\right)\partial_r (\delta E_t) - \frac{64b^2 k^2}{r^6 f}\delta E_t = 0. \tag{44}$$

### 4.1.1 General solutions

Let us define the following dimensionless combination of temperature and background magnetic field for later convenience:

$$\zeta(b/T^2) \equiv \frac{\sqrt{r_h^4 - 64b^2 k^2}}{r_h^2} = \left(\frac{b}{T^2 \pi^2}\right)\sqrt{\frac{\pi^4}{(b^2/T^4)} - 64k^2}. \tag{45}$$

In the small magnetic field limit (with $T$ fixed), that is $b \to 0$, we have,

$$\zeta(b/T^2) \underset{b\to 0}{\to} 1 - \frac{32k^2}{\pi^4}\left(\frac{b}{T^2}\right)^2 - \frac{512k^4}{\pi^8}\left(\frac{b}{T^2}\right)^4 + \mathcal{O}\left(\left(\frac{b}{T^2}\right)^6\right). \tag{46}$$

In the large magnetic field limit (with $T$ fixed), that is $b \to \infty$, we have,

$$\zeta(b/T^2) \underset{b\to\infty}{\to} \frac{8ik}{\pi^2}\left(\frac{b}{T^2}\right) - \frac{i\pi^2}{16k}\left(\frac{b}{T^2}\right)^{-1} - \frac{i\pi^6}{4096k^3}\left(\frac{b}{T^2}\right)^{-3} + \mathcal{O}\left(\left(\frac{b}{T^2}\right)^{-5}\right). \tag{47}$$

Notice from the definition of $\zeta$ (in Eq.(45)), it appears that $\zeta = 0$ could be a point of non-analyticity for the susceptibility $\chi(\zeta)$; we shall show below that $\chi$ is actually a function of $\zeta^2$ and not of $\zeta$ and hence is analytic at $\zeta = 0$.

---

[9]Note that we could have obtained Eq.(44) by directly taking $\omega \to 0$ limit in Eq.(43). This is because in $\omega \to 0$ limit $\delta E_r$ and $\delta E_t$ decouple.

Solving Eq.(44) analytically we find the general solution as,

$$\delta E_t(r)_{gen} = d_1\, r_h^{1+\zeta}\, r^{-1-\zeta}\, {}_2F_1\left(-\frac{1}{4}-\frac{\zeta}{4},\frac{1}{4}-\frac{\zeta}{4};1-\frac{\zeta}{2};\frac{r^4}{r_h^4}\right)$$
$$+ d_2\, r_h^{1-\zeta}\, r^{-1+\zeta}\, {}_2F_1\left(-\frac{1}{4}+\frac{\zeta}{4},\frac{1}{4}+\frac{\zeta}{4};1+\frac{\zeta}{2};\frac{r^4}{r_h^4}\right),\qquad(48)$$

where $d_1$ and $d_2$ are integration constants. From Eq.(48) we will fix them such that $\delta E_t(r)$ is regular near the horizon (or in the interior). The boundary condition we seek is $\delta E_t(r=r_h)_{gen}=0$. We define

$$\delta E_t(r,r_h,b,k,d_3)|_p \equiv r_h^{1+\zeta}\, r^{-1-\zeta}\, {}_2F_1\left(-\frac{1}{4}-\frac{\zeta}{4},\frac{1}{4}-\frac{\zeta}{4};1-\frac{\zeta}{2};\frac{r^4}{r_h^4}\right)$$
$$+ d_3\, r_h^{1-\zeta}\, r^{-1+\zeta}\, {}_2F_1\left(-\frac{1}{4}+\frac{\zeta}{4},\frac{1}{4}+\frac{\zeta}{4};1+\frac{\zeta}{2};\frac{r^4}{r_h^4}\right)\qquad(49)$$

$$\text{(such that } \delta(E_t)_{gen} = d_1\ \delta E_t(r,r_h,b,k,d_3)|_p \text{ and } d_3 := d_2/d_1\,).$$

Then we evaluate $\delta E_t(r=r_h,r_h,b,k,d_3)|_p$ at the horizon and find,

$$\delta E_t(r=r_h,r_h,b,k,d_3)|_p = \frac{\Gamma\left(1-\frac{\zeta}{2}\right)}{\Gamma\left(\frac{3}{4}-\frac{\zeta}{4}\right)\Gamma\left(\frac{5}{4}-\frac{\zeta}{4}\right)} + \frac{d_3\,\Gamma\left(1+\frac{\zeta}{2}\right)}{\Gamma\left(\frac{3}{4}+\frac{\zeta}{4}\right)\Gamma\left(\frac{5}{4}+\frac{\zeta}{4}\right)}.\qquad(50)$$

We further fix $d_3$ as

$$d_3(r_h,b,k) = -\frac{\Gamma\left(1-\frac{\zeta}{2}\right)\Gamma\left(\frac{3}{4}+\frac{\zeta}{4}\right)\Gamma\left(\frac{5}{4}+\frac{\zeta}{4}\right)}{\Gamma\left(1+\frac{\zeta}{2}\right)\Gamma\left(\frac{3}{4}-\frac{\zeta}{4}\right)\Gamma\left(\frac{5}{4}-\frac{\zeta}{4}\right)}.\qquad(51)$$

Note that, $d_3$ is chosen such a way that $\delta E_t(r=r_h,r_h,b,k,d_3)|_p = 0$ or in other words, $\delta E_t(r=r_h)_{gen}=0$.[10]

### 4.1.2 Regular solution

After $d_3$ is fixed as above we obtain the following regular solution,

$$\delta E_t(r) = r_h^{1-\zeta}\, r^{-1-\zeta}\, \Gamma\left(1-\frac{\zeta}{2}\right)\left[r_h^{2\zeta}\, {}_2\tilde{F}_1\left(-\frac{1}{4}-\frac{\zeta}{4},\frac{1}{4}-\frac{\zeta}{4};1-\frac{\zeta}{2};\frac{r^4}{r_h^4}\right)\right.$$
$$\left. -\, 2^{-\zeta}\, r^{2\zeta}\, \frac{\Gamma\left(\frac{3}{2}+\frac{\zeta}{2}\right)}{\Gamma\left(\frac{3}{2}-\frac{\zeta}{2}\right)}\, {}_2\tilde{F}_1\left(-\frac{1}{4}+\frac{\zeta}{4},\frac{1}{4}+\frac{\zeta}{4};1+\frac{\zeta}{2};\frac{r^4}{r_h^4}\right)\right],\qquad(52)$$

where ${}_2\tilde{F}_1$ is the regularized hypergeometric function.

Now we examine Eq.(52) in the vanishing magnetic field limit that is for $b=0$,

$$\delta E_t(r) \underset{b\to 0}{\to} -1 + \frac{r_h^2}{r^2} + \mathcal{O}\left(\frac{1}{r^2}\right).\qquad(53)$$

---

[10]Note that, $\delta E_t(r=r_h,r_h,b,k,d_3)|_p$ is a 'particular' solution (hence the notation $\delta E_t|_p$) which satisfies the boundary condition $\delta E_t(r=r_h)_{gen}=0$.

Now let us look at the boundary expansion of Eq.(52) $\left(\text{up to } \mathcal{O}\left(\frac{1}{r^2}\right)\right)$,

$$
\delta E_t(r) \underset{r \to \infty}{\to} \left(-\frac{1}{r_h^4}\right)^{\frac{1}{4}-\frac{\zeta}{4}} r_h^{1-\zeta} \left(\left(-\frac{1}{r_h^4}\right)^{\frac{\zeta}{2}} r_h^{2\zeta} - \tan\left(\left(\frac{1}{4}+\frac{\zeta}{4}\right)\pi\right)\right) \left(\frac{\sqrt{\pi}\,\Gamma\left(1-\frac{\zeta}{2}\right)}{\Gamma\left(\frac{1}{4}-\frac{\zeta}{4}\right)\Gamma\left(\frac{5}{4}-\frac{\zeta}{4}\right)}\right)
$$
$$
+ \frac{2}{r^2}\left(-\frac{1}{r_h^4}\right)^{\frac{3}{4}-\frac{\zeta}{4}} r_h^{5-\zeta} \left(\left(-\frac{1}{r_h^4}\right)^{\frac{\zeta}{2}} r_h^{2\zeta} - \cot\left(\left(\frac{1}{4}+\frac{\zeta}{4}\right)\pi\right)\right) \left(\frac{\sqrt{\pi}\,\Gamma\left(1-\frac{\zeta}{2}\right)}{\Gamma\left(-\frac{1}{4}-\frac{\zeta}{4}\right)\Gamma\left(\frac{3}{4}-\frac{\zeta}{4}\right)}\right).
$$
(54)

Notice that the above expression is of the form, $A + Br^{-2}$. From this we can find the charge susceptibility as $\chi = -\frac{2B}{A}$ (using the definition of the current from (21)).

$$
\chi(r_h, b) = -2r_h^2\left(\frac{\zeta^2-1}{16\pi^2}\cos\left(\frac{\zeta\pi}{2}\right)\Gamma^2\left(\frac{1-\zeta}{4}\right)\Gamma^2\left(\frac{1+\zeta}{4}\right)\right) = -2r_h^2\, g(\zeta) \equiv -2T^2\, \tilde{g}(b/T^2),
$$
(55)

where[11] $\tilde{g}(b/T^2) = \pi^2 g(\zeta) \equiv \frac{\zeta^2-1}{16}\cos\left(\frac{\zeta\pi}{2}\right)\Gamma^2\left(\frac{1-\zeta}{4}\right)\Gamma^2\left(\frac{1+\zeta}{4}\right)$. Note that $g(\zeta)$ is manifestly an even function of $\zeta$. Hence, $g(\zeta)$ is analytic as a function of $\zeta^2$ and not $\zeta$ which we wanted to show (see (45)), and $g(\zeta)$ is analytic at $\zeta = 0$ (see Appendix C for further details on the $\zeta = 0$ case).

Note that $\lim_{\zeta \to \pm 1} g(\zeta) = -1$. Furthermore, the vanishing magnetic field limit is $b = 0$ which corresponds to $\zeta = \pm 1$ (from the definition of $\zeta$). Hence, we find that in the vanishing magnetic field limit, $\chi(r_h, b = 0) = 2r_h^2$, which is the usual charge susceptibility for the conventional black brane, and matches with what we would have gotten from computing the susceptibility from Eq.(53)).

In the field-theoretical study [1], the corresponding susceptibility was taken to be the free fermion result at zero magnetic field $\chi_s = \frac{1}{6}T^2$. Let us contrast this to the susceptibility obtained above in Eq.(55) from holography. A key difference is that the proportionality factor relating $\chi$ and $T^2$ is no longer a constant but a function of $b/T^2$, namely $\tilde{g}(b/T^2)$. (Presumably a similar effect would exists in a perturbative approach, where one would simply consider the effects of Landau levels on the charge susceptibility).

A plot of $\chi/T^2$ as a function of $kb/T^2$ is shown in Figure 1.

## 5 Hydrodynamic limit

We now solve the bulk equations of motion in a small frequency limit; we will see an analogue of the chiral magnetic effect appear in this limit, and we will also reproduce from the bulk certain aspects of the hydrodynamic calculation in Section 2.

Let us begin by noting from (19) that the equation of motion for the 2-form $B_2$ can be written as

$$
\nabla_P \mathcal{H}^{PQR} = 0,
$$
(56)

where the 3-form $\mathcal{H}_3$ is defined as

$$
\mathcal{H}^{PQR} \equiv H^{PQR} + 2k\left[\epsilon^{PQRMN}E^L H_{LMN}\right] + \frac{2k}{3}\left[H_{LMN}\left(E^P\epsilon^{LMNQR} + E^Q\epsilon^{LMNRP} + E^R\epsilon^{LMNPQ}\right)\right],
$$
(57)

---

[11]We have used $\Gamma(1+x) = x\Gamma(x)$ and $\Gamma(x)\Gamma(1-x) = \frac{\pi}{\sin(\pi x)}$ to obtain Eq.(55) from Eq.(54).

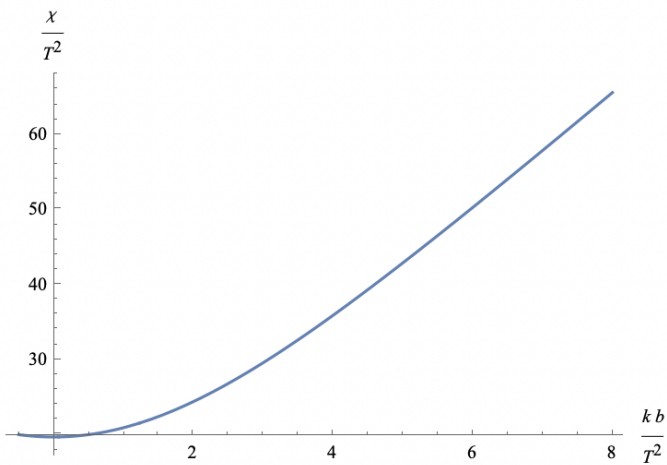

Figure 1: $\chi/T^2$ as a function of $kb/T^2$.

with $H_3 = dB_2$. This general form – i.e. that the equation of motion can be written as the divergence of a 3-form $\mathcal{H}_3$ – follows from the fact that the action is a function of $dB_2$ alone.

We are now interested in solving these equations of motion in a hydrodynamic limit, i.e. with $\frac{\omega}{T} \to 0$. When taking a small frequency limit in AdS/CFT, it is useful to use the formalism of the membrane paradigm [38]. The usual infalling membrane boundary condition as applied to the modified field-strength $\mathcal{H}$ results in the following condition at the black hole horizon:

$$\sqrt{-g}\,\mathcal{H}^{rxy}(r_h) = \mathcal{H}_{txy}(r_h)\Sigma(r_h), \qquad \Sigma(r) \equiv \sqrt{\frac{-g}{-g_{rr}g_{tt}}}\,g^{xx}g^{yy}\,. \tag{58}$$

See [18] for an application of these techniques to a minimally coupled 2-form $B_2$. Importantly, it is shown there that the quantity $\Sigma(r_h)$ can be understood as the conventional electric resistivity $\rho$.

We now study the consequences of this boundary condition for fluctuations about a background field configuration where $H_{rtz} \neq 0$ as in (40). We will study a configuration where the nonzero components of the fluctuations are $\mathcal{H}_{rxy}, \mathcal{H}_{txy}, E_t$ and $E_r$.

We begin by writing out:

$$\mathcal{H}_{txy} = H_{txy} + 8k\sqrt{-g}\,E_t H^{rtz} \tag{59}$$

(where we have used an orientation in which $\epsilon_{txyrz} < 0$). The boundary condition (58) thus implies that at the horizon we have

$$\sqrt{-g}\,\mathcal{H}^{rxy}(r_h) = \Sigma(r_h)\left(H_{txy} + 8k\sqrt{-g}\,E_t H^{rtz}\right)\Big|_{r=r_h}\,. \tag{60}$$

We would now like to propagate this information to the boundary, where it can be given an interpretation in the field theory. The equations of motion in the low-frequency limit take the form

$$\partial_r\left(\sqrt{-g}\,\mathcal{H}^{rxy}\right) = 0\,, \qquad \partial_r H_{txy} = 0\,, \tag{61}$$

where the former is the $xy$ component of the diagonal equation of motion (56) and the latter is the Bianchi identity associated with $H_3 = dB_2$. Thus we can evaluate the expression above at the AdS boundary:

$$\sqrt{-g}\,\mathcal{H}^{rxy}(\infty) = \Sigma(r_h)\left(H_{txy}(\infty) + 8k\sqrt{-g}\,E_t(r_h)H^{rtz}(r_h)\right)\,. \tag{62}$$

Now we note that $J^{xy} = -\lim_{r\to\infty} \sqrt{-g}\mathcal{H}^{rxy}$, and we thus find that

$$J^{xy} = -\Sigma(r_h)\big(H_{txy}(\infty) + 8k\sqrt{-g}E_t(r_h)H^{rtz}(r_h)\big). \tag{63}$$

Here $J^{xy}$ may be understood as the electric field in the $z$ direction $\mathcal{E}_z$; as explained before, $J^{tz} = -\sqrt{-g}H^{rtz}$ is the background magnetic field. Now $H_{txy}$ is the applied source; let us set it to zero. We then find the following expression:

$$J^{xy} = 8k\Sigma(r_h)E_t(r_h)J^{tz}. \tag{64}$$

Let us now pause to dissect this result. This appears reminiscent of the chiral magnetic effect; in the conventional language of electric and magnetic fields as in (3), $J^{xy}$ is proportional to the electric field in the $z$ direction; thus we see that in the absence of external sources, there is an electric field parallel to the applied magnetic field, where the constant of proportionality is $8k\Sigma(r_h)E_t(r_h)$. In comparing to field theory, we note that $\Sigma(r_h) = \rho$, i.e. the conventional electric resistivity in this theory.

Of course, if we are at precisely zero frequency, then we are required to have that $E_t(r_h) = 0$. This is completely consistent with the known physics of the chiral magnetic effect, which states that the *equilibrium* value of the chiral magnetic effect for the consistent vector current is zero [12].[12] This is thus an unexciting but expected result.

Now we should note however that in this work we are interested in small fluctuations around equilibrium. If $\omega \neq 0$, then it is no longer required that $E_t(r_h) = 0$ (indeed, in the conventional case of a massless gauge field, this quantity is no longer even gauge invariant). Let us instead allow $E_t(r) \neq 0$ and use the small frequency analysis above to compute the relaxation rate of the axial charge. Here we will make contact with the hydrodynamic calculation above, and we will thus study a situation where $E_t(\infty) = 0$, i.e. there is no axial source applied.

We first use (24) to write down

$$\partial_t j_A^t = 8kJ^{xy}J^{tz}. \tag{65}$$

This equation holds at all $r$, (indeed, from above, all of the expressions in it are radially constant), and so we can evaluate it at the boundary to find:

$$-i\omega j_A^t = 64k^2\Sigma(r_h)(J^{tz})^2 E_t(r_h). \tag{66}$$

Solving this for $\omega$ we find:

$$\omega = 64i\left(\frac{E_t(r_h)}{j_A^t}\right)\Sigma(r_h)\big(kJ^{tz}\big)^2. \tag{67}$$

Thus we see that there is a diffusion pole, where the coefficient of the pole varies as the magnetic field squared. We stress that the approximation made was $\omega \to 0$; from above we see that this also requires that $kJ^{tz} \to 0$. Away from that limit, we expect to see deviations from the quadratic expression above.

Let us also examine the pre-factor of the expression in the limit $kJ^{tz} \to 0$. We see that the ratio of $E_t(r_h)$ and $j_A^t$ appears. We can evaluate this from Eq.(53) (with the appropriate boundary conditions: $E_t(r_h) \neq 0$ and $E_t(\infty) = 0$) and Eq.(21) to get,

$$\frac{E_t(r_h)}{j_A^t} \xrightarrow{r\to\infty} -\frac{1}{2r_h^2} \equiv -\chi^{-1}. \tag{68}$$

---

[12]Here – as explained in detail in [12] – one must be careful about the distinction between the consistent and covariant currents; the covariant chiral magnetic effect is not zero, but the consistent one receives contributions both from the axial chemical potential and the value of the axial gauge field source, which precisely cancel in equilibrium.

We thus find (evaluating $\Sigma(r_h) = \frac{1}{r_h}$ from (39)):

$$\omega = -32ik^2 \left( \frac{b^2}{r_h^3} \right). \tag{69}$$

We should compare it to the expectation from elementary hydrodynamics given in (11). Putting in the holographic expressions for the field-theoretical quantities in (11) in the small magnetic field limit, using $\sigma = \frac{1}{\rho} = r_h$, and $\chi = 2r_h^2$, and $B = b$, we find that (11) becomes $\Gamma_A = \frac{32k^2b^2}{r_h^3}$, i.e. precisely the same as the pole exhibited above. This agreement is not surprising; indeed it can be seen that the derivation above parallels in the bulk the hydrodynamic calculation leading to (11).

However, here we see the limitations of the hydrodynamic calculation – in particular, we see explicitly in this holographic model that the analytic calculation is expected to break down if $kJ^{tz}$ is not small; i.e. the result above is valid only in the small $b$ limit. In the next section we explicitly compute the same relaxation rate numerically and compare with lattice results.

In this work we have neglected the backreaction of the charge degrees of freedom on the geometry. Our calculation is also entirely classical, in that we have ignored fluctuations, which in this framework are suppressed by $\frac{1}{N}$, with $N$ a proxy for the number of field-theoretical degrees of freedom. It is reasonable to ask whether such effects will change the picture above. As the actual low-frequency calculation in the bulk essentially exactly parallels the hydrodynamic calculation (given in Sec.2), it seems reasonable to expect that such corrections would change individually the values of things like $\Gamma_A$ and the resistivity $\rho$, but not change the *relationship* between them that we find here (see for instance Eq. (67)). This is broadly the expectation from the usual fluid-gravity correspondence. We note however that there are known examples in a similar hydrodynamic context where loop effects in the bulk can qualitatively change the infrared physics (see e.g. [39,40]). Generally such effects can be anticipated on field-theoretical grounds, and we return to this issue in the conclusion.

# 6 Numerical results

In this section we calculate the quasi-normal modes of our system using standard holographic techniques.

## 6.1 Contributing equations of motion

From here on we shall work in ingoing Eddington-Finkelstein coordinates $(r, v, x, y, z)$ rather than the Schwarzschild coordinates used above (see Appendix A for a brief review of the coordinate system). First let us give some useful expressions,

$$E^r = (E_v + r^2 f E_r), \qquad E^v = E_r,$$

$$H^{rxy} = \frac{f}{r^2} \partial_r (B_{xy}) - \frac{i\omega}{r^4} B_{xy}, \quad H^{vxy} = \frac{1}{r^4} \partial_r (B_{xy}).$$

We will study finite frequency fluctuations about the equilibrium solution (40). Consider $H \to H_0 + \delta H$ and $E \to E_0 + \delta E$ with $E_0 = 0$ and $H_0 = H^{rvz}$, which are the background solutions Eq.(40) (in the ingoing coordinates). $H_0$ and $E_0$ are the background fields and $\delta H$ and $\delta E$ are the fluctuations of the fields that we are interested in.

Before proceeding to solve Eq.(15) let us first note that we are interested in wave-like solutions for $\delta B$ and $\delta E$ of the form $e^{-i\omega v}$. So, we are considering the corresponding wave vector to be of the form $k^\mu = (\omega, \vec{k}) = (\omega, 0)$, i.e. the only non-zero momentum is in the

time direction. Furthermore, since the magnetic field is in the $z$-direction, the little group for fluctuations is $SO(2)$: the group of rotations in the $x - y$ plane. Therefore, we have the following contributing equations of motion in the relevant channels for the fluctuations: in the antisymmetric tensor channel (i.e. from the ($\rho\sigma = xy$) components of Eq.(15)),

$$-\left[\partial_\nu\left(\sqrt{-g}\,\delta H^{\nu xy}\right) + \partial_r\left(\sqrt{-g}\,\delta H^{rxy}\right)\right]$$
$$-8k\left[\partial_\nu\left(\sqrt{-g}\,\delta E^\nu \epsilon^{r\nu zxy}H_{r\nu z}\right) + \partial_r\left(\sqrt{-g}\,\delta E^r \epsilon^{r\nu zxy}H_{r\nu z}\right)\right] = 0 \quad (\rho\sigma = xy). \quad (70)$$

Similarly, from Eq.(16) we have

$$64k^2H^2\,\delta E^\nu - 4k\epsilon_{r\nu zxy}H^{r\nu z}\,\delta H^{\nu xy} = -\frac{1}{\sqrt{-g}}\partial_r\left(\sqrt{-g}\,\delta G^{r\nu}\right), \quad (71)$$

$$64k^2H^2\,\delta E^r - 4k\epsilon_{r\nu zxy}H^{r\nu z}\,\delta H^{rxy} = -\frac{1}{\sqrt{-g}}\partial_\nu\left(\sqrt{-g}\,\delta G^{\nu r}\right). \quad (72)$$

In the above contributing equations of motion, we have not considered terms with $\partial_x$, $\partial_y$, $\partial_z$ as we have set spatial momenta to vanish. The vector channel involving $\delta E^x$, $\delta H^{xyz}$ decouples and we will not consider it.

In these coordinates, the background solutions (40) become,

$$H^{r\nu z} = \frac{b}{r^3}, \qquad H_{r\nu z} = -\frac{b}{r}. \quad (73)$$

Now we solve for $\delta E^r$ in Eq.(72) to obtain,

$$\delta E_r = \frac{i\omega r^4\,\partial_r(\delta E_\nu) + 64k^2b^2\delta E_\nu + 4kbr^2f\,\partial_r(\delta B_{xy}) - 4kbi\omega\delta B_{xy}}{\tilde{\gamma}}, \quad (74)$$

where $\tilde{\gamma} := r^4\omega^2 - 64b^2k^2r^2f$.

Next we give Eqs.(70) and (71) in terms of the fields with all indices downstairs,

$$\partial_r^2\left(\delta B_{xy}\right)[-rf] + \partial_r\left(\delta B_{xy}\right)\left[-f - \frac{4r_h^4}{r^4} + \frac{2i\omega}{r}\right] + \delta B_{xy}\left[\frac{-i\omega}{r^2}\right] - \partial_r\left(\delta E_\nu\right)\left[\frac{8kb}{r}\right] + \delta E_\nu\left[\frac{8kb}{r^2}\right]$$

$$-\partial_r\left(\delta E_r\right)[8kbrf] + \delta E_r\left[-8kbf - \frac{32kbr_h^4}{r^4} + \frac{8kbi\omega}{r}\right] = 0, \quad (75)$$

$$\frac{-64b^2k^2}{r^4}\delta E_r - \frac{4kb}{r^4}\partial_r\left(\delta B_{xy}\right) = \partial_r^2\left(\delta E_\nu\right) + \left(\frac{3}{r}\right)\partial_r\left(\delta E_\nu\right) + i\omega\partial_r\left(\delta E_r\right) + \left(\frac{3}{r}\right)i\omega\delta E_r. \quad (76)$$

If we substitute $\delta E_r$ from Eq.(74) into Eqs.(75) and (76), then we obtain two coupled ODEs – these are somewhat lengthy so we do not present them explicitly, but we solve them numerically below.

## 6.2 Numerics

Now we shall solve Eq.(75) and Eq.(76) numerically using a mid-point shooting[13] method (see e.g. [41] for a discussion of the shooting method, with some previous applications to quasinormal modes in [42,43]). Below we present some details of the boundary conditions; the reader interested only in the results can feel free to skip to the next section.

---

[13]In mid-point shooting method we numerically integrate the boundary solution from boundary and horizon solution from horizon and adjust these till they meet somewhere in the middle and this adjustment yields the QNM.

### 6.2.1 Logarithmic fall-off

$B_{xy}$ has a logarithmic fall-off near the boundary, associated with the fact that the double-trace deformation associated with $J^2$ on the boundary is marginally (ir)relevant [18]. As explained in detail in that work, the correct boundary condition at the UV cut-off $u = u_\Lambda$ takes the form:

$$B_{xy}(u_\Lambda) - \frac{J}{\kappa} = 0,\tag{77}$$

where $J = u\partial_u B_{xy}$ and $\kappa$ the double-trace coupling for $J^2$. The form of $B_{xy}$ we take is,

$$B_{xy}(u) = d_0 + \sum_j d_j\, u^j + \ln(u)\left[d_0' + \sum_j d_j'\, u^j\right],\tag{78}$$

where the $d_i$ are expansion coefficients. Using (77) we then find:

$$d_0 + \sum_j d_j\, u^j + \ln(u_\Lambda)\left[d_0' + \sum_j d_j'\, u^j\right] - \frac{1}{\kappa}\left[\sum_j j d_j\, u^j + d_0' + \sum_j d_j'\, u^j + \sum_j j\, u^j \ln(z) d_j'\right] = 0.$$

Now at $u = u_\Lambda$, (79) becomes (for $u \to 0$),

$$d_0 + \ln(u_\Lambda)d_0' - \frac{d_0'}{\kappa} = 0,$$
$$d_0 = d_0'\left[\ln\left(e^{\frac{1}{\kappa}}/u_\Lambda\right)\right].\tag{79}$$

Thus, at the boundary ($u = 0$) we obtain $B_{xy}$ as,

$$B_{xy}(u) = d_0'\ln\left(u e^{\frac{1}{\kappa}}/u_\Lambda\right) + \sum_j d_j\, u^j + (\ln(u))\sum_j d_j'\, u^j,$$
$$B_{xy}(u) = d_0'\ln\left(u/u^*\right) + \sum_j d_j\, u^j + (\ln(u))\sum_j d_j'\, u^j,\tag{80}$$

where $u^* = u_\Lambda e^{-\frac{1}{\kappa}}$ is an RG-invariant combination of the double-trace coupling and the UV cut-off; this is the analogue of the Landau pole in regular QED, and by dimensional transmutation all physical results can depend on this alone (see [18, 44] for a discussion in the holographic context).

Now since $B_{xy}$ at the boundary has a logarithmic fall-off, the coupled nature of the equations of motion as given in Eq.(75) and Eq.(76) imply that $E_v$ has the following form at the boundary:

$$E_v(u) = u^2\left(c_0 + \sum_j c_j\, u^j\right) + \ln(u)\left(\sum_j cc_j\, u^j\right).\tag{81}$$

The logarithm appearing in this boundary condition appears to follow from the fact that the axial current $j_A^\mu$ mixes with the 2-form current $J^{\mu\nu}$.

Next we present the numerical results; see Appendix D for further details on the numerical implementation.

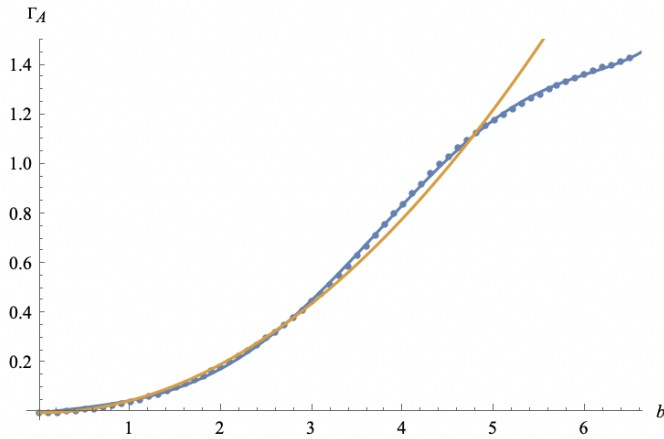

Figure 2: $\Gamma_A$ vs $b$ with $k = 0.0375$ and $r_h = 1$ (blue curve is numerics, orange curve is quadratic fit).

### 6.2.2 Hydrodynamic mode

The lowest quasinormal mode $\omega_l = -i\Gamma_A$ approaches the origin as $b$ approaches zero. For small $b$ (i.e. for $0 < b \leq 3$) it varies with $b$ as

$$\Gamma_A(r_h, b) = 0.048 \left( \frac{b^2}{r_h^3} \right), \tag{82}$$

where the prefactor was obtained from a numerical fit, and where we have restored $r_h$ on dimensional grounds.

Next we move onto some higher values of $b$ to show that away from a small neighbourhood of $b = 0$; the quadratic $b^2$ behaviour of no longer captures the full dependence and we obtain a more complicated function of $b$. We consider $0 < b \leq 6.5$ and obtain the behaviour shown in Fig.(2), where the orange curve is as given in Eq.(82). Note that both the curves above match till about $b \approx 3$.

Now let us compare the above result obtained from numerics to the result obtained in Section 5 using the membrane paradigm formalism (in the $kJ^{tz} \to 0$ limit) in Eq.(69). Note that if we put $k = 0.0375$ in Eq.(69) then we find,

$$\Gamma_A(r_h, b) = 0.045 \left( \frac{b^2}{r_h^3} \right), \tag{83}$$

in approximate agreement (within 6%) with the small-frequency limit of the numerics. As mentioned in that section, the membrane paradigm analysis also agrees with elementary hydrodynamics arguments arising from treating electrodynamics perturbatively; thus we conclude that at small $b$ the conventional chiral MHD approach from weakly gauged electrodynamics is valid.

However, from Fig.(2) we notice that for $b \gg 1$, the functional dependence of $\Gamma_A$ on $b$ is no longer quadratic, and is a non-trivial function of $b$. This function now appears to depend on UV physics, and is not simply determined by other thermodynamic quantities such as the susceptibility $\chi$.

For example, we can try to improve the hydrodynamic result for $\Gamma_A$ in (11) with the holographically determined susceptibility in (55). The resulting plot as a function of the magnetic field $B = b$ is shown in Figure 3. It appears barely different from the quadratic dependence as

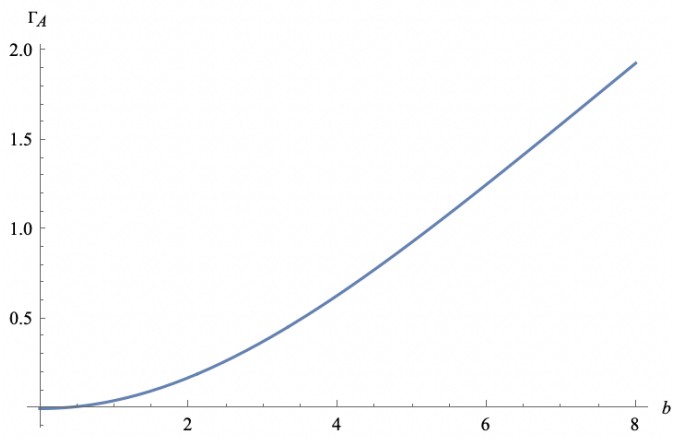

Figure 3: $\Gamma_A^{\text{improved}}$ from (11) (i.e. chiral MHD with weakly coupled electrodynamics) as a function of $b$ with $k = 0.0375$ and $r_h = 1$; note it does not capture the non-trivial dependence on $b$ seen in the numerical results of Figure 2.

$\chi$ does not depend strongly on $b$. Indeed if we expand this improved hydrodynamic attempt at approximating $\Gamma_A^{\text{improved}}$ as a series in $b$ we get,

$$\Gamma_A^{\text{improved}}(b) = 32\,k^2\left(\frac{b^2}{r_h^3}\right) + \mathcal{O}(b^4) = 0.045\,b^2 + \mathcal{O}(b^4), \tag{84}$$

where in the second equality above we have put $k = 0.0375$ and $r_h = 1$ (for a comparison with the numerical parameters) and the coefficient of $\mathcal{O}(b^2)$ is 100 times that of the coefficient of $\mathcal{O}(b^4)$. Thus the dependence of the charge susceptibility on $b$ is insufficient to account for the non-trivial dependence of $\Gamma_A(b)$, and it appears to not be determined by thermodynamic data.

Finally, in our numerical investigation we also observed many non-hydrodynamic gapped modes. These do not seem to have model-independent relevance, but we discuss them in Appendix E.

## 7 Conclusion

In this work we have discussed a holographic model that is in the same universality class as a massless Dirac fermion coupled to QED at finite temperature, i.e. where axial charge is non-conserved due to an anomaly with a dynamical operator (involving a topological density constructed from the 2-form current associated with magnetic flux conservation) on the right hand side. We described the bulk dualization process by which we constructed the holographic model and computed some basic observables.

Perhaps our most significant result was explicit computation of the axial charge relaxation rate $\Gamma_A$ in the presence of a background magnetic field; we found that due to the anomaly the axial charge density $j_A^t$ is not conserved, and instead obeys an equation of the form $j_A^t \sim e^{-\Gamma_A t}$, where $\Gamma_A$ was numerically found through solving the bulk equations of motion. It is a nontrivial function of the background magnetic field $B$ and the temperature $T$, and can be seen in Figure 2. Note that at small magnetic field this relaxation rate is quadratic in the field $B$; indeed as the pole approaches the origin the pre-factor may be computed analytically from the small frequency limit of the bulk equations of motion. This pre-factor can also be obtained from elementary magneto-hydrodynamic arguments that essentially treat the anomaly coefficient

perturbatively, as reviewed in Section 2. The resulting pre-factor agrees with our holographic computation, and we thus find that at small magnetic fields:

$$\Gamma_A^{\text{hol}}(B \to 0) \approx \gamma^{\text{MHD}} B^2. \tag{85}$$

At larger magnetic fields this is a non-trivial function of $B$ and can only be obtained from a full holographic treatment.[14]

We now discuss the previous literature on this result. A lattice study of a field-theoretical model in the same universality class was recently performed in [1,2]; in particular, [1] studied the diffusion of the operator $F \wedge F$ (which can be related to the charge relaxation rate by a fluctuation-dissipation argument), and [2] directly measured $\Gamma_A$. Those works numerically observe an expression of the form:

$$\Gamma_A^{\text{lattice}}(B) \approx \gamma^{\text{lattice}} B^2. \tag{86}$$

Interestingly, those works found that $\gamma^{\text{lattice}}/\gamma^{\text{MHD}} \approx 10$, i.e. that the pre-factor obtained from the lattice differs from the hydrodynamic estimate by an order of magnitude [2]. In those works it was argued that this means that short-distance physics that is not taken into account in the hydrodynamic analysis is important. Interestingly, this is not what we find from our (UV-complete) holographic calculation; instead our holographic result precisely coincides with the hydrodynamic result at small magnetic fields, differing from it only at larger fields when the magnetic field *itself* probes UV scales.

It is interesting to speculate on the cause of this discrepancy between the lattice and holography. An ingredient entering into the computation of $\gamma^{\text{MHD}}$ is the resistivity of the electromagnetic sector; in holography it is very easy to see how this enters into the calculation and separate it from the anomalous dynamics but in a purely field-theoretical treatment it seems possible that uncertainties in this conductivity – a notoriously complicated quantity to calculate from first principles – could cloud this analysis, as was already suggested in [2]. It would be interesting to perform further tests of this hypothesis, perhaps by computing more observables from holography and the lattice and comparing them further.

If we take our results at face value, it suggests that for this observable, a hydrodynamic treatment of conventional MHD (treating the anomaly as a perturbation) is sufficient at weak magnetic fields, though it differs quantitatively from the true result at stronger fields.[15]

There are many directions for future research. Our bulk action Eq.(37) permits the explicit study of a strongly interacting system in the same universality class as the chiral plasma. It would be very interesting to understand other phenomena, e.g. if the instabilities (due to non-vanishing $\vec{k}$) [21] exist in this model, or the study of chiral magnetic waves [46].

From a field-theoretical point of view, it would be very interesting to go further than our holographic considerations and construct a true effective hydrodynamic theory for this system. Indeed this was one of the motivations for our construction of the holographic action (37), though it is sufficiently complicated that it does not shed much immediate light on how (or whether) an effective description could be computed. Indeed, the prospect of such an analysis is clouded by the fact that we are not aware of a completely universal field-theoretical description of the anomaly (1); conventional lore would tell us the axial symmetry is simply completely broken, though as we have argued this appears to miss important physics associated with the fact that it is broken not by a generic operator but rather by a topological

---

[14]We note that Eq.(11) states that the relaxation rate vanishes in the limit of vanishing magnetic field. It is at the moment not clear to us whether this is an artifact of the classical description; for example it is possible that when one includes fluctuations there is a non-vanishing relaxation rate even at zero magnetic field. In principle this could be evaluated using an appropriate Kubo formula of the topological density; we leave this for further investigation, and we thank Luca Delacrétaz for this comment.

[15]This is philosophically aligned with previous results [45] that argue that various transport coefficients are generically renormalized if the gauge fields sourcing the anomaly are dynamical.

density $J \wedge J$ constructed from the current of a 1-form symmetry. Indeed, along these lines recent work describes a novel higher group structure that is present when the axial symmetry is spontaneously broken [47–49]. It would be very interesting to understand whether such an analysis could be extended to the phase when the axial symmetry is unbroken or realized at finite temperature, as a first step towards constructing a hydrodynamic EFT.

**Note added:** Shortly after the first version of this paper appeared on the arXiv, two papers [22, 23] appeared where a precise characterization of the ABJ anomaly is given in terms of non-invertible symmetries. It would be very interesting to use this refined understanding to construct a hydrodynamic theory.

## Acknowledgments

This work was supported in part by the STFC Consolidated Grant ST/P000371/1 (NI & RG). We would like to express our gratitude to Daniele Dorigoni, Tin Sulejmanpasic, Stefano Cremonesi, Iñaki García Etxebarria and Aristomenis Donos for valuable discussions on gauge transformations, monopoles and hydrodynamics. We would like to thank Umut Gursoy, Diego Hofman, Luca Delacrétaz, Kieran Macfarlane, Napat Poovuttikul, Adrien Florio, Zheng Liang Lim, Gabriel Arenas-Henriquez, Richie Dadhley, T.V. Karthik and Swagat Saurav Mishra for useful discussions on related issues.

## A   Conventions

### A.1   Units and conventions regarding differential forms

We work in natural units with $c = \hbar = 1$ and our metric signature is mostly plus: $(-, +, +, \cdots \cdots, +)$.

For boundary indices we have $\mu \nu \rho \sigma \lambda \cdots$ and for bulk indices we have $MNPQR \cdots$. We have for the epsilon symbol, $\tilde{\epsilon}_{0123\cdots} = +1$ and for the volume form, $\epsilon = +\sqrt{-g} \, d^D x$, in $D$ spacetime dimensions.

In ingoing EF coordinates $(r, v, x, y, z)$ we have,

$$ds_5^2 = -r^2 f(r) dv^2 + 2 dv dr + r^2 \left( dx^2 + dy^2 + dz^2 \right), \tag{A.1}$$

$$\sqrt{-g} = r^3, \tag{A.2}$$

$$\epsilon_{rvzxy} = r^3 \tilde{\epsilon}_{rvzxy} = r^3, \tag{A.3}$$

$$\epsilon^{rvzxy} = -r^{-3}, \tag{A.4}$$

where $f \equiv f(r) := \left( 1 - \frac{r_h^4}{r^4} \right)$ and $\partial_r f(r) = \frac{4 r_h^4}{r^5}$.

We also record a few useful identities relating differential forms to their components, starting with:

$$A_p \wedge \star A_p = \frac{1}{p!} A_{\mu_1 \cdots \mu_p} A^{\mu_2 \cdots \mu_p} \epsilon. \tag{A.5}$$

Integrating we find,

$$\int_{\mathcal{M}_D} A_p \wedge \star A_p = \frac{1}{p!} \int_{\mathcal{M}_D} d^D x \sqrt{-g} A_{\mu_1 \cdots \mu_p} A^{\mu_1 \cdots \mu_p}. \tag{A.6}$$

We also sometimes use expressions of the following form,

$$\int_{\mathcal{M}_5} H_3 \wedge F_2 = \frac{(-1)^s}{2!3!} \int_{\mathcal{M}_5} d^5x \sqrt{-g} \epsilon^{\mu\nu\rho\alpha\beta} H_{\mu\nu\rho} F_{\alpha\beta} , \tag{A.7}$$

where $s$ is the number of minus signs in the metric.

## A.2 Conventions regarding definition of the boundary current

For this let us consider free Maxwell action in $D$ spacetime (bulk) dimensions as given below,

$$S = \frac{1}{2} \int_{\mathcal{M}_D} F_{p+1} \wedge \star F_{p+1} = \frac{1}{2} \int_{\mathcal{M}_D} \frac{1}{(p+1)!} \left(F_{p+1}\right)^2 , \tag{A.8}$$

where $\left(F_{p+1}\right)^2 \equiv F^{\mu_1...\mu_{p+1}} F_{\mu_1...\mu_{p+1}}$. The boundary dual of the above theory has a magnetic $(D-p-3)$-form symmetry, $J_{D-p-2} \equiv \star F_{p+1}\big|_{r\to\infty}$. So, in $D = 5$ spacetime (bulk) dimensions we have the usual story that $J_2 \equiv \star F_2\big|_{r\to\infty}$ (with $r$ being the holographic radial coordinate).

Now let us Poincaré dualize the above action to get,

$$S_{\text{dual}} = \frac{1}{2} \int_{\mathcal{M}_D} H_{D-p-1} \wedge \star H_{D-p-1} = \frac{1}{2} \int_{\mathcal{M}_D} \frac{1}{(D-p-1)!} \left(H_{D-p-1}\right)^2 , \tag{A.9}$$

where, $H_{D-p-1} = dB_{D-p-2}$ with $B_{D-p-2}$ being the Lagrange multiplier to enforce the closure of $F_{p+1}$ during the dualization procedure. The dualization gives, $H_{D-p-1} = \star F_{p+1}$ which in turn implies that,

$$J_{D-p-2} = H_{D-p-1}\big|_{r\to\infty} . \tag{A.10}$$

The AdS/CFT dictionary we need to define the boundary current is,

$$\left\langle \exp\left(\frac{1}{p!} \int b_{\mu_1...\mu_p} J^{\mu_1...\mu_p}\right) \right\rangle_{\text{CFT}} = \mathcal{Z}_{\text{grav}}\left[B_{\mu_1...\mu_p}(r\to\infty) = b_{\mu_1...\mu_p}\right],$$

leading to, $\quad J^{\mu_1...\mu_p} = p! \lim_{r\to\infty} \frac{\delta S_{\text{bulk}}}{\delta\left(\partial_r B_{\mu_1...\mu_p}\right)} . \tag{A.11}$

Note that there is a factor of $p!$ in the definition of the boundary current in A.11. This factor is needed to get A.10. Let us show this below.

Now let us obtain the boundary current from $S_{\text{dual}}$ using A.11.

$$\frac{\delta S}{\delta\left(\partial_r B_{\mu_1...\mu_{D-p-2}}\right)} = \frac{1}{2(D-p-1)!} \frac{\partial \left(H_{D-p-1}\right)^2}{\partial\left(\partial_r B_{\mu_1...\mu_{D-p-2}}\right)} = \frac{1}{2(D-p-1)!} 2H^{r\mu_2...\mu_{D-p-2}}(D-p-1)$$

$$= \frac{1}{(D-p-2)!} H^{r\mu_2...\mu_{D-p-1}} . \tag{A.12}$$

Thus, we see that to get A.10, we should have the following normalization in the boundary current definition,

$$J^{\mu_2...\mu_{D-p-1}} = (D-p-2)! \lim_{r\to\infty} \frac{\delta S_{\text{bulk}}}{\delta\left(\partial_r B_{\mu_2...\mu_{D-p-1}}\right)} . \tag{A.13}$$

So, A.13 explains the factor of $p!$ in A.11.

# B  Inverse operation

In terms of tensor-index notation, Eq.(30) is,

$$F_{MN} = -\frac{1}{2}\epsilon_{PQRMN}\,(\partial^P B^{QR}) + 4k\,\epsilon_{PQRMN}\,A^P F^{QR}\,,$$

$$\text{leading to, } \left[\delta_M^Q \delta_N^R - 4k\,\epsilon_{IJKMN}\,g^{JQ}g^{KR}A^I\right]F_{QR} = -\frac{1}{2}\epsilon_{IJKMN}\,\partial^I(B^{JK})\,, \qquad \text{(B.1)}$$

$$\text{leading to, } \mathcal{O}^{QR}_{\ \ MN}\,F_{QR} = -\frac{1}{2}\epsilon_{IJKMN}\,\partial^I(B^{JK})\,, \qquad \text{(B.2)}$$

where $\mathcal{O}^{QR}_{\ \ MN} \equiv \delta_M^Q \delta_N^R - 4k\,\epsilon_{IJKMN}\,g^{JQ}g^{KR}A^I$.

Now the task is to invert $\mathcal{O}^{QR}_{\ \ MN}$ to obtain $(\mathcal{O}^{-1})^{MN}_{\ \ LK}$ such that $\mathcal{O}^{QR}_{\ \ MN}\,(\mathcal{O}^{-1})^{MN}_{\ \ LK} = \delta_L^Q \delta_K^R$.[16] Then, $(\mathcal{O}^{-1})^{MN}_{\ \ LK}$ would enable us to write $F_2$ in terms of $B_2$ and $A_1$ and their derivatives which is what we are after.

Note that we are not assuming that $\mathcal{O}^{QR}_{\ \ MN}$ or $(\mathcal{O}^{-1})^{MN}_{\ \ LK}$ is anti-symmetric at this point. However, ultimately they will be contracted with $F_{QR}$ (for instance, see Eq.(B.1)), and their symmetric parts would cancel out and things will turn out to be consistent.

Let us consider the most general $(\mathcal{O}^{-1})^{MN}_{\ \ LK}$ possible (arranged in ascending powers of $A_1$) and then we shall demand it to be $\mathcal{O}^{QR}_{\ \ MN}$'s inverse. The most general expression for $(\mathcal{O}^{-1})^{MN}_{\ \ LK}$ is,

$$\begin{aligned}
(\mathcal{O}^{-1})^{MN}_{\ \ LK} &= c_0\,\delta_L^M \delta_K^N + \bar{c}_0\,\delta_K^M \delta_L^N + 4c_1 k\,\epsilon_{PIJLK}\,A^P\,g^{IM}g^{JN} + 16c_2 k^2 \delta_L^M A^N A_K \\
&\quad + 16\bar{c}_2 k^2 \delta_L^N A^M A_K + 16c_2' k^2 \delta_K^N A^M A_L + 16\tilde{c}_2 k^2 \delta_K^M A^N A_L\,, \qquad \text{(B.3)}
\end{aligned}$$

where $c_0$, $\bar{c}_0$, $c_1$, $c_2$, $\bar{c}_2$, $c_2'$, $\tilde{c}_2$ are coefficients to be determined by demanding that $(\mathcal{O}^{-1})^{MN}_{\ \ LK}$ is the inverse of $\mathcal{O}^{QR}_{\ \ MN}$ (their subscript is numbered as per the powers of $A_1$ they are coefficients of). Note that we cannot have any more powers of $A_1$ in the above expression (in the sense of anti-symmetric indices of $A_1$) as when they would be contracted with $\epsilon_{PIJQR}$ they would cancel. Note that, every term other than terms whose coefficients are $c_0$ and $\tilde{c}_0$ have to come with some powers of $k$ otherwise on $k \to 0$ limit they would not give the proper inverse of the $\mathcal{O}^{QR}_{\ \ MN}\big|_{k\to 0} = \delta_M^Q \delta_N^R$ as they would survive the $k \to 0$ limit.

Now demanding, $\mathcal{O}^{QR}_{\ \ MN}\,(\mathcal{O}^{-1})^{MN}_{\ \ LK} = \delta_L^Q \delta_K^R$ we get the following equation,

$$\begin{aligned}
&(c_0 + 32c_1 k^2 A^2)\delta_L^Q \delta_K^R + (\bar{c}_0 - 32c_1 k^2 A^2)\delta_K^Q \delta_L^R + (c_1 - c_0 + \bar{c}_0)4k\,\epsilon_{PIJLK}\,A^P\,g^{IQ}g^{JR} \\
&+ (c_2 - 2c_1)16k^2\,\delta_L^Q A^R A_K + (\bar{c}_2 + 2c_1)16k^2\,\delta_L^R A^Q A_K + (c_2' - 2c_1)16k^2\,\delta_K^R A^Q A_L \\
&+ (\tilde{c}_2 + 2c_1)16k^2\,\delta_K^Q A^R A_L = \delta_L^Q \delta_K^R\,.
\end{aligned}$$

So the above equation is satisfied if,

$$c_1 = \frac{1}{1 + 64k^2 A^2}\,, \quad c_0 = \frac{1 + 32k^2 A^2}{1 + 64k^2 A^2}\,, \quad \bar{c}_0 = \frac{32k^2 A^2}{1 + 64k^2 A^2}\,, \qquad \text{(B.4)}$$

$$c_2 = c_2' = 2c_1 = \frac{2}{1 + 64k^2 A^2}\,, \quad \bar{c}_2 = \tilde{c}_2 = -2c_1 = \frac{-2}{1 + 64k^2 A^2}\,. \qquad \text{(B.5)}$$

One can readily check that with the above values for the coefficients $(\mathcal{O}^{-1})^{MN}_{\ \ LK}$ as given in Eq.(B.3) is the inverse of $\mathcal{O}^{QR}_{\ \ MN}$ (and also gives the correct inverse in the $k \to 0$ limit when $c_0 \to 1$ and only the term with $c_0$ as the coefficient survives).

Next we multiply, Eq.(B.2) with $(\mathcal{O}^{-1})^{MN}_{\ \ LK}$ to get Eq.(36),

---

[16]Note that $\delta_L^Q \delta_K^R F_{QR} = F_{LK}$. So, in the space of 2 forms $\delta_L^Q \delta_K^R$ is the identity operator.

For completeness here we express the differential forms that make up $S_{5p}$ in terms of their components (up to quadratic orders in $E$),

$$-\frac{1}{2}F_2 \wedge \star F_2 \rightarrow \left[\frac{H^2}{12} - 48k^2(E\cdot H)^2 + \frac{32}{3}k^2H^2E^2 + \frac{2k}{3}\epsilon_{PQRLK}H^{PQR}E_MH^{MLK}\right]\tilde{c}_1^2, \quad \text{(B.6)}$$

$$+H_3 \wedge F_2 \rightarrow \left[-\frac{H^2}{6} - \frac{64}{3}k^2H^2E^2 - \frac{2k}{3}\epsilon_{PQRLK}H^{PQR}E_MH^{MLK} + 32k^2(E\cdot H)^2\right]\tilde{c}_1^2, \quad \text{(B.7)}$$

$$-4kE_1 \wedge F_2 \wedge F_2 \rightarrow \left[32k^2(E\cdot H)^2 - \frac{k}{3}\epsilon_{PQRLK}H^{PQR}E_MH^{MLK}\right]\tilde{c}_1^2, \quad \text{(B.8)}$$

$$-\frac{1}{2}G_2 \wedge \star G_2 \rightarrow -\frac{1}{4}G^2, \quad \text{(B.9)}$$

and when expanded in powers of small $k$, $\tilde{c}_1^2 = 1 - 128\,k^2E^2 + \mathcal{O}(k^4)$.

## C  $\zeta \rightarrow 0$ and hypergeometric differential equation

Note that if we naively put $\zeta = 0$ in (48), then we do not get two linearly independent solutions at $\zeta = 0$. This is related to the structure of the Riemann differential equation about the point $\zeta = 0$. Here we give the general solution to Eq.(44) in the limit of $\zeta \rightarrow 0$,[17]

$$\delta E_t(r)_{gen}\Big|_{\zeta\rightarrow 0} = \frac{r_h}{r}\left[m_1\,_2F_1\left(-\frac{1}{4},\frac{1}{4};1;\frac{r^4}{r_h^4}\right) + m_2\left\{_2F_1\left(-\frac{1}{4},\frac{1}{4};1;\frac{r^4}{r_h^4}\right)\ln\left(\frac{r^4}{r_h^4}\right)\right.\right.$$

$$\left.\left. + \sum_{i=0}^{\infty}\frac{(a)_i(b)_i}{(c)_i\,i!}\left(\frac{r^4}{r_h^4}\right)^i\left(\psi\left(i - \frac{1}{4}\right) + \psi\left(i + \frac{1}{4}\right) - 2\psi(1 + i)\right)\right\}\right], \quad \text{(C.1)}$$

where $a = -\frac{1}{4}$, $b = \frac{1}{4}$, $c = 1$, $\psi(x) \equiv \frac{d}{dx}\ln(\Gamma(x))$ is the digamma function, $m_1$ and $m_2$ are integration constants, and $(a)_i$ is the rising Pochhammer symbol defined as,

$$(a)_i := \begin{cases} 1, & i = 0, \\ a(a+1)\cdots\cdots(a+i-1), & i > 0. \end{cases}$$

## D  Implementation of numerics

In this appendix we aim to find a searching condition for the numerical implementation of our quasinormal modes. To do this we perform a simple linear algebra exercise. Suppose we have $N$ fields $\Phi^I(r)$. Let $\Phi_a^I(r)$ be a basis for the solutions that are ingoing at horizon ($a \in \{1,\cdots\cdots,N\}$). Let $\Phi_\alpha^I(r)$ be a basis for the solutions that are outgoing at horizon ($\alpha \in \{1,\cdots\cdots,N\}$). Let $r^*$ be the matching point. We wish to know when $\exists\ C_a$ and $D_\alpha$ such that,

$$\sum_a C_a\Phi_a^I(r^*) = \sum_\alpha D_\alpha\Phi_\alpha^I(r^*), \quad \text{(D.1)}$$

$$\sum_a C_a\Phi_a^{'I}(r^*) = \sum_\alpha D_\alpha\Phi_\alpha^{'I}(r^*), \quad \text{(D.2)}$$

---

[17]For more information see section 15.10 of [50].

which implies that there exists a solution that satisfies both sets of boundary conditions.

We can frame the above as a linear algebra problem. Consider $X_a^A := \left\{ \Phi_a^I(r^*), \Phi_a^{'I}(r^*) \right\}$ (with $A \in \{1, \cdots\cdots, 2N\}$) and view $X_a^A$ as a subspace of $\mathbb{R}^{2N}$. Similarly, consider $Y_\alpha^A := \left\{ \Phi_\alpha^I(r^*), \Phi_\alpha^{'I}(r^*) \right\}$. We wish to know when the subspaces $X_a^A$ and $Y_\alpha^A$ overlap. This condition will then imply the existence of a solution that would satisfy both sets of boundary conditions (D.1) and (D.2). Let us consider $\left(Y^\perp\right)_b^A$ that satisfies,

$$\sum_A \left(Y^\perp\right)_b^A Y_\alpha^A = 0, \tag{D.3}$$

where $b \in \{1, \cdots\cdots, N\}$.

Then we want,

$$\sum_A \left(Y^\perp\right)_b^A X_a^A = 0, \tag{D.4}$$

to have a non-trivial solution. Eq.(D.4) will have a non-trivial solution if and only if,

$$\det_{a,b} \left[ \sum_A \left(Y^\perp\right)_b^A X_a^A \right] = 0. \tag{D.5}$$

Eq.(D.5) is the QNM searching condition that we have been looking for. Next we perform the numerics[18] with the mid-point shooting method. For the matching point, we have, $y_m = 0.6$. The numerical parameters used are,

| Tolerance ($t_m$) | Horizon radius ($r_h$) | UV cut-off ($u_\Lambda$) |
|---|---|---|
| 0.1 | 1 | 0.1 |
| Double-trace coupling ($\kappa$) | RG parameter ($u^*$) | Anomaly coefficient ($k$) |
| $-1/\ln(10)$ | 1 | 0.0375 |

# E  Non-hydrodynamic (gapped) modes and quasinormal mode table

We observe from the numerics that, there exists a higher (generically) complex non-hydrodynamic mode $\forall\ b \in [0.00001, 20]$. This mode seems to be independent of $b$ as it exists $\forall\ b \in [0.00001, 20]$ and has the value,

$$\Gamma_{\text{non-hydro},0}^{\text{cplx}} = \pm 0.965 - 1.736i. \tag{E.1}$$

However, $\forall\ b \in (0, 15.3]$, $\Gamma_{\text{non-hydro},0}^{\text{cplx}}$ is not the lowest QNM and for $\forall\ b \geq 15.4$, $\Gamma_{\text{non-hydro},0}^{\text{cplx}}$ becomes the lowest QNM.

---

[18]An order of convergence of $10^{-11}$ and lower has been treated as zero in the numerics. The numerical results that are presented have been verified (up to very slight variations) for the matching point in the range $y_m \in [0.2, 0.8]$. We have also dropped a few terms in the UV and the IR expansions of the fields and have verified the robustness of the numerical results.

We also give below some more generically complex non-hydro (gapped) modes which exist $\forall\, b \in [0.00001, 20]$,

$$\Gamma^{\text{cplx}}_{\text{non-hydro},1} = \pm 3.359 - 3.697i\,, \tag{E.2}$$

$$\Gamma^{\text{cplx}}_{\text{non-hydro},2} = \pm 5.334 - 5.663i\,, \tag{E.3}$$

$$\Gamma^{\text{cplx}}_{\text{non-hydro},3} = \pm 7.175 - 6.797i\,, \tag{E.4}$$

$$\Gamma^{\text{cplx}}_{\text{non-hydro},4} = \pm 8.96 - 8.604i\,, \tag{E.5}$$

$$\Gamma^{\text{cplx}}_{\text{non-hydro},5} = \pm 10.537 - 7.337i\,, \tag{E.6}$$

$$\Gamma^{\text{cplx}}_{\text{non-hydro},6} = \pm 13.102 - 5.463i\,. \tag{E.7}$$

Table 1: Lowest QNM vs $b$.

| S.No | $b$ | $-i\Gamma_A$ |
|---|---|---|
| 0 | $10^{-5}$ | $-2.3721962 \times 10^{-12}\,i$ |
| 1 | 0.1 | $-0.000450031\,i$ |
| 2 | 0.2 | $-0.0018005\,i$ |
| 3 | 0.3 | $-0.00405253\,i$ |
| 4 | 0.4 | $-0.00720806\,i$ |
| 5 | 0.5 | $-0.0112698\,i$ |
| 6 | 0.6 | $-0.0162414\,i$ |
| 7 | 0.7 | $-0.0221275\,i$ |
| 8 | 0.8 | $-0.0289337\,i$ |
| 9 | 0.9 | $-0.036667\,i$ |
| 10 | 1.0 | $-0.0453354\,i$ |
| 11 | 1.1 | $-0.0549485\,i$ |
| 12 | 1.2 | $-0.0655176\,i$ |
| 13 | 1.3 | $-0.0770557\,i$ |
| 14 | 1.4 | $-0.0895777\,i$ |
| 15 | 1.5 | $-0.103101\,i$ |
| 16 | 1.6 | $-0.117644\,i$ |
| 17 | 1.7 | $-0.13323\,i$ |
| 18 | 1.8 | $-0.149884\,i$ |
| 19 | 1.9 | $-0.167633\,i$ |
| 20 | 2.0 | $-0.186508\,i$ |
| 21 | 2.1 | $-0.206544\,i$ |
| 22 | 2.2 | $-0.227778\,i$ |
| 23 | 2.3 | $-0.250252\,i$ |
| 24 | 2.4 | $-0.27401\,i$ |
| 25 | 2.5 | $-0.299101\,i$ |
| 26 | 2.6 | $-0.325573\,i$ |
| 27 | 2.7 | $-0.353477\,i$ |
| 28 | 2.8 | $-0.382864\,i$ |
| 29 | 2.9 | $-0.41378\,i$ |
| 30 | 3.0 | $-0.446263\,i$ |
| 31 | 3.1 | $-0.480341\,i$ |
| 32 | 3.2 | $-0.516016\,i$ |
| 33 | 3.3 | $-0.553261\,i$ |

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
