# Peer review of "Higher-form symmetries, anomalous magnetohydrodynamics, and holography"

_SciPost Physics, doi:SciPost Phys. 14, 163 (2023)_

## Round 1 · Referee Report · Luca Delacrétaz · 2022-11-22

Report

The authors have addressed my questions.

A loophole remain concerning the longevity of the axial current, which they argue for in their reply:

"[...] this analysis appears to show that at small magnetic fields the decay rate vanishes as B^2 (where B is the magnetic field). If this is correct, it can indeed be made parametrically small, and one might seek a universal description that works at least in this regime"

However they also acknowledge that there could be a universal B independent contribution to $\Gamma_A$ coming from hydrodynamic fluctuations. I agree with the authors that this contribution may turn out to vanish ("It also seems possible that the appropriate low-frequency correlator always vanishes due to special properties of the topological density"), but it would be nice to confirm this with the explicit calculation. I leave it to the authors whether they prefer to include this calculation in this paper or save it for future work.

I recommend this paper for publication.

---

## Round 1 · Author Response

We have replied to the referees in the Reply to a Report page and hence we do not repeat our replies here.

---

## Round 1 · List of Changes

List of changes:

1. Added footnote [2] on page 8 commenting on the bulk holographic action in Eq.(3.2)

2. Changed the definition of susceptibility in Eq.(4.3) and added how susceptibility is related to the chiral chemical potential in the linear regime in Eq.(4.4)

3. Added a paragraph above Numerical Results’ section explaining issues related to back-reaction and 1/N effects which we elaborate upon in the report

4. Added footnote [14] below Eq.(7.1) to comment on the scenario where the relaxation rate may not vanish even if the magnetic field vanishes. We elaborate upon this further in the report.

---

## Editorial Decision

published